# Can Class-Priors Help Single-Positive Multi-Label Learning?

**Biao Liu, Ning Xu*, Jie Wang, Xin Geng**
School of Computer Science and Engineering, Southeast University,
Nanjing 210096, China
Key Laboratory of New Generation Artificial Intelligence Technology and
Its Interdisciplinary Applications (Southeast University),
Ministry of Education, China
*Corresponding author: xning@seu.edu.cn

## Abstract

Single-positive multi-label learning (SPMLL) is a weakly supervised multi-label learning problem, where each training example is annotated with only one positive label. Existing SPMLL methods typically assign pseudo-labels to unannotated labels with the assumption that prior probabilities of all classes are identical. However, the class-prior of each category may differ significantly in real-world scenarios, which makes the predictive model not perform as well as expected due to the unrealistic assumption on real-world application. To alleviate this issue, a novel framework named CRISP, i.e., Class-pRiors Induced Single-Positive multi-label learning, is proposed. Specifically, a class-priors estimator is introduced, which can estimate the class-priors that are theoretically guaranteed to converge to the ground-truth class-priors. In addition, based on the estimated class-priors, an unbiased risk estimator for classification is derived, and the corresponding risk minimizer can be guaranteed to approximately converge to the optimal risk minimizer on fully supervised data. Experimental results on ten MLL benchmark datasets demonstrate the effectiveness and superiority of our method over existing SPMLL approaches.

## 1 Introduction

Multi-label learning (MLL) is a learning paradigm that aims to train a model on examples associated with multiple labels to accurately predict relevant labels for unknown instances [43, 25]. Over the past decade, MLL has been successfully applied to various real-world applications, including image annotation [30], text classification [24], and facial expression recognition [2].

Compared with multi-class-single-label learning, where each example is associated with a unique label, MLL involves instances that are assigned multiple labels. As the number of examples or categories is large, accurately annotating each label of an example becomes exceedingly challenging. To address the high annotation cost, single-positive multi-label learning (SPMLL) has been proposed [5, 38], where each training example is annotated with only one positive label. Moreover, since many examples in multi-class datasets, such as ImageNet [42], contain multiple categories but are annotated with a single label, employing SPMLL allows for the derivation of multi-label predictors from existing numerous multi-class datasets, thereby expanding the applicability of MLL.

To address the issue that model tends to predict all labels as positive if trained with only positive labels, existing SPMLL methods typically assign pseudo-labels to unannotated labels. Cole et al. updates the pseudo-labels as learnable parameters with a regularization to constrain the number of expected positive labels [5]. Xu et al. recovers latent soft pseudo-labels by employing variational label enhancement [38]. Zhou et al. adopts asymmetric-tolerance strategies to update pseudo-labels

cooperating with an entropy-maximization loss [45]. Xie et al. utilizes contrastive learning to learn the manifold structure information and updates the pseudo-labels with a threshold [33].

These approaches rely on a crucial assumption that prior probabilities of all classes are identical. However, in real-world scenarios, the class-prior of each category may differ significantly. This unrealistic assumption will introduce severe biases into the pseudo-labels, further impacting the training of the model supervised by the inaccurate pseudo-labels. As a result, the learned model could not perform as well as expected.

Motivated by the above consideration, we propose a novel framework named CRISP, i.e., Class-pRiors Induced Single-Positive multi-label learning. Specifically, a class-priors estimator is derived, which determines an optimal threshold by estimating the ratio between the fraction of positive labeled samples and the total number of samples receiving scores above the threshold. The estimated class-priors can be theoretically guaranteed to converge to the ground-truth class-priors. In addition, based on the estimated class-priors, an unbiased risk estimator for classification is derived, which guarantees the learning consistency [26] and ensures that the obtained risk minimizer would approximately converge to the optimal risk minimizer on fully supervised data. Our contributions can be summarized as follows:

- Practically, for the first time, we propose a novel framework for SPMLL named CRISP, which estimates the class-priors and then an unbiased risk estimator is derived based on the estimated class-priors, addressing the unrealistic assumption of identical class-priors for all classes.

- Theoretically, the estimated class-priors can be guaranteed to converge to the ground-truth class-priors. Additionally, we prove that the risk minimizer corresponding to the proposed risk estimator can be guaranteed to approximately converge to the optimal risk minimizer on fully supervised data.

Experiments on four multi-label image classification (MLIC) datasets and six MLL datasets show the effectiveness of our methods over several existing SPMLL approaches.

## 2   Related Work

Multi-label learning is a supervised machine learning technique where an instance is associated with multiple labels simultaneously. The study of label correlations in multi-label learning has been extensive, and these correlations can be categorized into first-order, second-order, and high-order correlations. First-order correlations involve adapting binary classification algorithms for multi-label learning, such as treating each label as an independent binary classification problem [1, 27]. Second-order correlations model pairwise relationships between labels [7, 9]. High-order correlations take into account the relationships among multiple labels, such as employing graph convolutional neural networks to extract correlation information among all label nodes [3]. Furthermore, there has been an increasing interest in utilizing label-specific features, which are tailored to capture the attributes of a specific label and enhance the performance of the models [41, 11].

In practice, accurately annotating each label for every instance in multi-label learning is unfeasible due to the immense scale of the output space. Consequently, multi-label learning with missing labels (MLML) has been introduced [28]. MLML methods primarily rely on low-rank, embedding, and graph-based models. The presence of label correlations implies a low-rank output space [25], which has been extensively employed to fill in the missing entries in a label matrix [35, 40, 34]. Another widespread approach is based on embedding techniques that map label vectors to a low-dimensional space, where features and labels are jointly embedded to exploit the complementarity between the feature and label spaces [39, 31]. Additionally, graph-based models are prevalent solutions for MLML, constructing a label-specific graph for each label from a feature-induced similarity graph and incorporating manifold regularization into the empirical risk minimization framework [28, 32].

In SPMLL, a specific case of multi-label learning with incomplete labels, only one of the multiple positive labels is observed. The initial work treats all unannotated labels as negative and updates the pseudo-labels as learnable parameters, applying a regularization to constrain the number of expected positive labels [5]. A label enhancement process [37, 22, 21, 36, 16] is used to recover latent soft labels and train the multi-label classifier [38]. The introduction of an asymmetric pseudo-label approach utilizes asymmetric-tolerance strategies for pseudo-labels, along with an entropy-maximization loss

[45]. Additionally, Xie et al. proposes a label-aware global consistency regularization method, leveraging the manifold structure information learned from contrastive learning to update pseudo-labels [33]. Liu et al. investigates the theoretical guarantee of pseudo-label-based methods for SPMLL [23], proving the learnability of such methods and proposing a mutual label enhancement approach that iteratively refines the label distributions [15, 17, 19, 18, 14] of samples and optimizes the multi-label classifier.

# 3 Preliminaries

## 3.1 Multi-Label Learning

Let $\mathcal{X} = \mathbb{R}^q$ denote the instance space and $\mathcal{Y} = \{0, 1\}^c$ denote the label space with $c$ classes. Given the MLL training set $\mathcal{D} = \{(\boldsymbol{x}_i, \boldsymbol{y}_i) | 1 \leq i \leq n\}$ where $\boldsymbol{x}_i \in \mathcal{X}$ is a $q$-dimensional instance and $\boldsymbol{y}_i \in \mathcal{Y}$ is its corresponding labels. Here, $\boldsymbol{y}_i = [y_i^1, y_i^2, \ldots, y_i^c]$ where $y_i^j = 1$ indicates that the $j$-th label is a relevant label associated with $\boldsymbol{x}_i$ and $y_i^j = 0$ indicates that the $j$-th label is irrelevant to $\boldsymbol{x}_i$. Multi-label learning is intended to produce a multi-label classifier in the hypothesis space $h \in \mathcal{H} : \mathcal{X} \mapsto \mathcal{Y}$ that minimizes the following classification risk:

$$\mathcal{R}(h) = \mathbb{E}_{(\boldsymbol{x}, \boldsymbol{y}) \sim p(\boldsymbol{x}, \boldsymbol{y})} \left[ \mathcal{L}(h(\boldsymbol{x}), \boldsymbol{y}) \right], \tag{1}$$

where $\mathcal{L} : \mathcal{X} \times \mathcal{Y} \mapsto \mathbb{R}^+$ is a multi-label loss function that measures the accuracy of the model in fitting the data. Note that a method is risk-consistent if the method possesses a classification risk estimator that is equivalent to $\mathcal{R}(f)$ given the same classifier [26].

## 3.2 Single-Positive Multi-Label Learning

For single-positive multi-label learning (SPMLL), each instance is annotated with only one positive label. Given the SPMLL training set $\tilde{\mathcal{D}} = \{(\boldsymbol{x}_i, \gamma_i) | 1 \leq i \leq n\}$ where $\gamma_i \in \{1, 2, \ldots, c\}$ denotes the only observed single positive label of $\boldsymbol{x}_i$. For each SPMLL training example $(\boldsymbol{x}_i, \gamma_i)$, we use the observed single-positive label vector $\boldsymbol{l}_i = [l_i^1, l_i^2, \ldots, l_i^c]^\top \in \{0, 1\}^c$ to represent whether $j$-th label is the observed positive label, i.e., $l_i^j = 1$ if $j = \gamma_i$, otherwise $l_i^j = 0$. The task of SPMLL is to induce a multi-label classifier $h \in \mathcal{H} : \mathcal{X} \mapsto \mathcal{Y}$ from $\tilde{\mathcal{D}}$, which can assign the unknown instance with a set of relevant labels.

# 4 The Proposed Method

## 4.1 The CRISP Algorithm

In this section, we introduce our novel framework, CRISP, i.e., Class-pRiors Induced Single-Positive multi-label learning. This framework alternates between estimating class-priors and optimizing an unbiased risk estimator under the guidance of the estimated class-priors.

Firstly, we introduce the class-priors estimator for SPMLL, leveraging the blackbox classifier $f$ to estimate the class-prior of each label. The class-priors estimator exploits the classifier $f$ to give each input a score, indicating the likelihood of it belonging to a positive sample of $j$-th label. Specifically, the class-priors estimator determines an optimal threshold by estimating the ratio between the fraction of the total number of samples and that of positive labeled samples receiving scores above the threshold, thereby obtaining the class-prior probability of the $j$-th label.

Motivated by the definition of top bin in learning from positive and unlabeled data (PU learning) [10], for a given probability density function $p(\boldsymbol{x})$ and a classifier $f$, define the threshold cumulative density function $q_j(z) = \int_{S_z} p(\boldsymbol{x}) d\boldsymbol{x}$ where $S_z = \{\boldsymbol{x} \in \mathcal{X} : f^j(\boldsymbol{x}) \geq z\}$ for all $z \in [0, 1]$. $q_j(z)$ captures the cumulative density of the feature points which are assigned a value larger than a threshold $z$ by the classifier of the $j$-th label. We now define an empirical estimator of $q_j(z)$ as $\hat{q}_j(z) = \frac{1}{n} \sum_{i=1}^n \mathbf{1}(f^j(\boldsymbol{x}_i) \geq z)$ where $\mathbf{1}(\cdot)$ is the indicator function. For each probability density function $p_j^p = p(\boldsymbol{x} | y_j = 1)$, $p_j^n = p(\boldsymbol{x} | y_j = 0)$ and $p = p(\boldsymbol{x})$, we define $q_j^p = \int_{S_z} p(\boldsymbol{x} | y_j = 1) d\boldsymbol{x}$ and $q_j^n = \int_{S_z} p(\boldsymbol{x} | y_j = 0) d\boldsymbol{x}$ respectively.

**Algorithm 1** CRISP Algorithm

---

**Input:** The SPMLL training set $\tilde{\mathcal{D}} = \{(\boldsymbol{x}_i, \gamma_i)|1 \leq i \leq n\}$, the multi-label classifier $f$, the number of epoch $T$, hyperparameters $0 \leq \delta, \tau \leq 1$;
1: **for** $t = 1$ **to** $T$ **do**
2:    **for** $j = 1$ **to** $c$ **do**
3:       Extract the positive-labeled samples set $\mathcal{S}_{L_j} = \{\boldsymbol{x}_i : l_i^j = 1, 1 \leq i \leq n\}$.
4:       Estimate $\hat{q}_j(z) = \frac{1}{n}\sum_{i=1}^{n} \mathbf{1}(f^j(\boldsymbol{x}_i) \geq z)$ and $\hat{q}_j^p(z) = \frac{1}{n_j^p}\sum_{\boldsymbol{x} \in \mathcal{S}_{L_j}} \mathbf{1}(f^j(\boldsymbol{x}) \geq z)$ for all $z \in [0, 1]$.
5:       Estimate the class-prior of $j$-th label by $\hat{\pi}_j = \frac{\hat{q}_j(\hat{z})}{\hat{q}_j^p(\hat{z})}$ with the threshold induced by Eq. (2).
6:    **end for**
7:    Update the model $f$ by forward computation and back-propagation by Eq. (7) using the estimated class-priors.
8: **end for**
**Output:** The predictive model $f$.

---

The steps involved in the procedure are as follows: Firstly, for each label, we extract a positive-labeled samples set $\mathcal{S}_{L_j} = \{\boldsymbol{x}_i : l_i^j = 1, 1 \leq i \leq n\}$ from the entire dataset. Next, with $\mathcal{S}_{L_j}$, we estimate the fraction of the total number of samples that receive scores above the threshold $\hat{q}_j(z) = \frac{1}{n}\sum_{i=1}^{n} \mathbf{1}(f^j(\boldsymbol{x}_i) \geq z)$ and that of positive labeled samples receiving scores above the threshold $\hat{q}_j^p(z) = \frac{1}{n_j^p}\sum_{\boldsymbol{x} \in \mathcal{S}_{L_j}} \mathbf{1}(f^j(\boldsymbol{x}) \geq z)$ for all $z \in [0, 1]$, where $n_j^p = |\mathcal{S}_{L_j}|$ is the cardinality of the positive-labeled samples set of $j$-th label. Finally, the class-prior of $j$-th label is estimated by $\frac{\hat{q}_j(\hat{z})}{\hat{q}_j^p(\hat{z})}$ at $\hat{z}$ that minimizes the upper confidence bound defined in Theorem 4.1.

**Theorem 4.1.** *Define* $z^\star = \arg\min_{z\in[0,1]} q_j^n(z)/q_j^p(z)$, *for every* $0 < \delta < 1$, *define* $\hat{z} = \arg\min_{z\in[0,1]} \left( \frac{\hat{q}_j(z)}{\hat{q}_j^p(z)} + \frac{1+\tau}{\hat{q}_j^p(z)} \left( \sqrt{\frac{\log(4/\delta)}{2n}} + \sqrt{\frac{\log(4/\delta)}{2n_j^p}} \right) \right)$. *Assume* $n_j^p \geq 2\frac{\log 4/\delta}{q_j^p(z^\star)}$, *the estimated class-prior* $\hat{\pi}_j = \frac{\hat{q}_j(\hat{z})}{\hat{q}_j^p(\hat{z})}$ *satisfies with probability at least* $1 - \delta$:

$$\pi_j - \frac{c_1}{q_j^p(z^\star)} \left( \sqrt{\frac{\log(4/\delta)}{2n}} + \sqrt{\frac{\log(4/\delta)}{2n_p}} \right) \leq \hat{\pi}_j \leq \pi_j + (1 - \pi_j)\frac{q_j^n(z^\star)}{q_j^p(z^\star)} +$$

$$\frac{c_2}{q_j^p(z^\star)} \left( \sqrt{\frac{\log(4/\delta)}{2n}} + \sqrt{\frac{\log(4/\delta)}{2n_p}} \right),$$

where $c_1, c_2 \geq 0$ are constants and $\tau$ is a fixed parameter ranging in $(0, 1)$. The proof can be found in Appendix A.1. Theorem 4.1 provides a principle for finding the optimal threshold. Under the condition that the threshold $\hat{z}$ satisfies:

$$\hat{z} = \arg\min_{z\in[0,1]} \left( \frac{\hat{q}_j(z)}{\hat{q}_j^p(z)} + \frac{1+\tau}{\hat{q}_j^p(z)} \left( \sqrt{\frac{\log(4/\delta)}{2n}} + \sqrt{\frac{\log(4/\delta)}{2n_j^p}} \right) \right), \tag{2}$$

the estimated class-prior $\hat{\pi}_j$ of $j$-th category will converge to the ground-truth class-prior with enough training samples. Practically, to determine the optimal threshold in Eq. (2), we conduct an exhaustive search across the set of outputs generated by the function $f^j$ for each class. The details can be found in Appendix A.2.

After obtaining an accurate estimate of class-prior for each category, we proceed to utilize these estimates as a form of supervision to guide the training of our model. Firstly, the classification risk $\mathcal{R}(f)$ on fully supervised information can be written as [1]:

$$\mathcal{R}(f) = \mathbb{E}_{(\boldsymbol{x},\boldsymbol{y})\sim p(\boldsymbol{x},\boldsymbol{y})} \left[ \mathcal{L}(f(\boldsymbol{x}), \boldsymbol{y}) \right] = \sum_{\boldsymbol{y}} p(\boldsymbol{y})\mathbb{E}_{\boldsymbol{x}\sim p(\boldsymbol{x}|\boldsymbol{y})} \left[ \mathcal{L}(f(\boldsymbol{x}), \boldsymbol{y}) \right]. \tag{3}$$

---

[1]The datail is provided in Appendix A.3.

In Eq. (3), the loss function $\mathcal{L}(f(\boldsymbol{x}), \boldsymbol{y})$ is calculated for each label separately, which is a commonly used approach in multi-label learning:

$$\mathcal{L}(f(\boldsymbol{x}), \boldsymbol{y}) = \sum_{j=1}^{c} y_j \ell(f^j(\boldsymbol{x}), 1) + (1 - y_j)\ell(f^j(\boldsymbol{x}), 0). \tag{4}$$

By substituting Eq. (4) into Eq. (3), the classification risk $\mathcal{R}(f)$ can be written as follows with the absolute loss function[2]:

$$\mathcal{R}(f) = \sum_{j=1}^{c} 2p(y_j = 1)\mathbb{E}_{\boldsymbol{x} \sim p(\boldsymbol{x}|y_j=1)} \left[1 - f^j(\boldsymbol{x})\right] + \left(\mathbb{E}_{\boldsymbol{x} \sim p(\boldsymbol{x})} \left[f^j(\boldsymbol{x})\right] - p(y_j = 1)\right). \tag{5}$$

The rewritten classification risk comprises two distinct components. The first component computes the risk solely for the positively labeled samples, and the second component leverages the unlabeled data to estimate difference between the expected output of the model $f$ and the class-prior $\pi_j = p(y_j = 1)$ to align the expected class-prior outputted by model with the ground-truth class-prior.

During the training process, the prediction of model can be unstable due to insufficiently labeled data. This instability may cause a large divergence between the expected class-prior $\mathbb{E}[f^j(\boldsymbol{x})]$ and the ground-truth class-prior $\pi_j$, even leading to a situation where the difference between $\mathbb{E}[f^j(\boldsymbol{x})]$ and $\pi_j$ turns negative [44]. To ensure non-negativity of the classification risk and the alignment of class-priors, absolute function is added to the second term. Then the risk estimator can be written as:

$$\mathcal{R}_{sp}(f) = \sum_{j=1}^{c} 2\pi_j \mathbb{E}_{\boldsymbol{x} \sim p(\boldsymbol{x}|y_j=1)} \left[1 - f^j(\boldsymbol{x})\right] + \left| \mathbb{E}_{\boldsymbol{x} \sim p(\boldsymbol{x})} \left[f^j(\boldsymbol{x})\right] - \pi_j \right|. \tag{6}$$

Therefore, we could express the empirical risk estimator via:

$$\widehat{\mathcal{R}}_{sp}(f) = \sum_{j=1}^{c} \frac{2\pi_j}{|\mathcal{S}_{L_j}|} \sum_{\boldsymbol{x} \in \mathcal{S}_{L_j}} \left(1 - f^j(\boldsymbol{x})\right) + \left| \frac{1}{n} \sum_{\boldsymbol{x} \in \tilde{\mathcal{D}}} \left(f^j(\boldsymbol{x}) - \pi_j\right) \right|. \tag{7}$$

The proposed equation enables the decomposition of the risk over the entire dataset into terms that can be estimated using both labeled positive and unlabeled samples.

In MLL datasets, where the number of negative samples for each label significantly exceeds that of positive samples, there is a tendency for the decision boundary to be biased towards the center of positive samples, especially for rare classes. This bias is further exacerbated in SPMLL due to the common strategy of assuming unobserved labels as negative [5, 38, 33] to warm up the model. To alleviate the issue, we propose a modification of Eq. (7):

$$\widehat{\mathcal{R}}_{sp}(f) = \sum_{j=1}^{c} \frac{2\pi_j}{|\mathcal{S}_{L_j}|} \sum_{\boldsymbol{x} \in \mathcal{S}_{L_j}} \left(1 - \frac{1}{1 + e^{-(g^j(\boldsymbol{x}) + \lambda \pi^j)}}\right) + \left| \frac{1}{n} \sum_{\boldsymbol{x} \in \tilde{\mathcal{D}}} \left(f^j(\boldsymbol{x}) - \pi_j\right) \right|. \tag{8}$$

where $\lambda$ is a hyper-parameter and $g^j(\boldsymbol{x})$ represents the logit of $j$-th label outputted by the network for instance $\boldsymbol{x}$ and $f^j(\boldsymbol{x}) = \sigma(g^j(\boldsymbol{x}))$ where $\sigma(\cdot)$ denotes the sigmoid function.

The algorithmic description of CRISP is shown in Algorithm 1.

## 4.2 Estimation Error Bound

In this subsection, an estimation error bound is established for Eq. (7) to demonstrate its learning consistency. Firstly, we define the function spaces as:

$$\mathcal{G}_{sp}^{L} = \left\{(\boldsymbol{x}, \boldsymbol{l}) \mapsto \sum_{j=1}^{c} 2\pi_j l_j \left(1 - f^j(\boldsymbol{x})\right) | f \in \mathcal{F}\right\}, \mathcal{G}_{sp}^{U} = \left\{(\boldsymbol{x}, \boldsymbol{l}) \mapsto \sum_{j=1}^{c} \left(f^j(\boldsymbol{x}) - \pi_j\right) | f \in \mathcal{F}\right\},$$

and denote the expected Rademacher complexity [26] of the function spaces as:

$$\widetilde{\mathfrak{R}}_n \left(\mathcal{G}_{sp}^{L}\right) = \mathbb{E}_{\boldsymbol{x}, y, \boldsymbol{\sigma}} \left[\sup_{g \in \mathcal{G}_{sp}^{L}} \sum_{i=1}^{n} \sigma_i g\left(\boldsymbol{x}_i, y_i\right)\right], \widetilde{\mathfrak{R}}_n \left(\mathcal{G}_{sp}^{U}\right) = \mathbb{E}_{\boldsymbol{x}, y, \boldsymbol{\sigma}} \left[\sup_{g \in \mathcal{G}_{sp}^{U}} \sum_{i=1}^{n} \sigma_i g\left(\boldsymbol{x}_i, y_i\right)\right],$$

Table 1: Predictive performance of each comparing method on four MLIC datasets in terms of *mean average precision (mAP)* (mean ± std). The best performance is highlighted in bold (the larger the better).

| | VOC | COCO | NUS | CUB |
|---|---|---|---|---|
| AN | 85.546±0.294 | 64.326±0.204 | 42.494±0.338 | 18.656±0.090 |
| AN-LS | 87.548±0.137 | 67.074±0.196 | 43.616±0.342 | 16.446±0.269 |
| WAN | 87.138±0.240 | 65.552±0.171 | 45.785±0.192 | 14.622±1.300 |
| EPR | 85.228±0.444 | 63.604±0.249 | 45.240±0.338 | 19.842±0.423 |
| ROLE | 88.088±0.167 | 67.022±0.141 | 41.949±0.205 | 14.798±0.613 |
| EM | 88.674±0.077 | 70.636±0.094 | 47.254±0.297 | 20.692±0.527 |
| EM-APL | 88.860±0.080 | 70.758±0.215 | 47.778±0.181 | 21.202±0.792 |
| SMILE | 87.314±0.150 | 70.431±0.213 | 47.241±0.172 | 18.611±0.144 |
| PLC | 88.021±0.121 | 70.422±0.062 | 46.211±0.155 | **21.840±0.237** |
| LL-R | 87.784±0.063 | 70.078±0.008 | 48.048±0.074 | 18.966±0.022 |
| LL-CP | 87.466±0.031 | 70.460±0.032 | 48.000±0.077 | 19.310±0.164 |
| LL-CT | 87.054±0.214 | 70.384±0.058 | 47.930±0.010 | 19.012±0.097 |
| BOOSTLU+LL-R | 89.224±0.017 | 73.272±0.006 | 49.590±0.021 | 19.136±0.009 |
| BOOSTLU+LL-CP | 88.358±0.212 | 70.820±0.030 | 47.810±0.166 | 18.166±0.063 |
| BOOSTLU+LL-CT | 88.528±0.053 | 71.742±0.006 | 48.216±0.021 | 17.952±0.007 |
| CRISP | **89.820±0.191** | **74.640±0.219** | **49.996±0.316** | 21.650±0.178 |

where $\boldsymbol{\sigma} = \{\sigma_1, \sigma_2, \cdots, \sigma_n\}$ is $n$ Rademacher variables with $\sigma_i$ independently uniform variable taking value in $\{+1, -1\}$. Then we have:

**Theorem 4.2.** *Assume the loss function* $\mathcal{L}_{sp}^L = \sum_{j=1}^c 2\pi_j l_j \left(1 - f^j(\boldsymbol{x})\right)$ *and* $\mathcal{L}_{sp}^U = \sum_{j=1}^c \left(f^j(\boldsymbol{x}) - \pi_j\right)$ *could be bounded by* $M$, *i.e.,* $M = \sup_{\boldsymbol{x} \in \mathcal{X}, f \in \mathcal{F}, \boldsymbol{y} \in \mathcal{Y}} \max(\mathcal{L}_{sp}^L(f(\boldsymbol{x}), \boldsymbol{y}), \mathcal{L}_{sp}^U(f(\boldsymbol{x}), \boldsymbol{y}))$, *with probability at least* $1 - \delta$, *we have:*

$$\mathcal{R}(\hat{f}_{sp}) - \mathcal{R}(f^\star) \leq \frac{4\sqrt{2}\rho}{C} \sum_{j=1}^c \mathfrak{R}_n(\mathcal{H}_j) + \frac{M}{\min_j |\mathcal{S}_{L_j}|} \sqrt{\frac{\log \frac{4}{\delta}}{2n}} + 4\sqrt{2} \sum_{j=1}^c \mathfrak{R}_n(\mathcal{H}_j) + M \sqrt{\frac{\log \frac{4}{\delta}}{2n}}.$$

where $C$ is a constant, $\hat{f}_{sp} = \min_{f \in \mathcal{F}} \widehat{\mathcal{R}}_{sp}(f)$, $f^\star = \min_{f \in \mathcal{F}} \mathcal{R}(f)$ are the empirical risk minimizer and the true risk minimizer respectively and $\rho = \max_j 2\pi_j$, $\mathcal{H}_j = \left\{h : \boldsymbol{x} \mapsto f^j(\boldsymbol{x}) | f \in \mathcal{F}\right\}$ and $\mathfrak{R}_n(\mathcal{H}_j) = \mathbb{E}_{p(\boldsymbol{x})}\mathbb{E}_{\boldsymbol{\sigma}} \left[\sup_{h \in \mathcal{H}_j} \frac{1}{n} \sum_{i=1}^n h(\boldsymbol{x}_i)\right]$. The proof can be found in Appendix A.5.

Theorem 4.2 shows that, as $n \to \infty$, $\hat{f}_{sp}$ would converge to $f^\star$ with an intrinsic error quantified by the Rademacher complexity terms, reflecting the complexity of the hypothesis space. Note that the error is a fundamental aspect of the learning problem and remains even in a fully supervised scenario [26].

## 5 Experiments

### 5.1 Experimental Configurations

**Datasets.** In the experimental section, our proposed method is evaluated on four large-scale multi-label image classification (MLIC) datasets and six widely-used multi-label learning (MLL) datasets. The four MLIC datasets include PSACAL VOC 2021 (VOC) [8], MS-COCO 2014 (COCO) [20], NUS-WIDE (NUS) [4], and CUB-200 2011 (CUB) [29]; the MLL datasets cover a wide range of scenarios with heterogeneous multi-label characteristics. For each MLIC dataset, 20% of the training set is withheld for validation. Each MLL dataset is partitioned into train/validation/test sets at a ratio of 80%/10%/10%. One positive label is randomly selected for each training instance, while the validation and test sets remain fully labeled. Detailed information regarding these datasets can be found in Appendix A.7. *Mean average precision (mAP)* is utilized for the four MLIC datasets [5, 33, 45] and five popular multi-label metrics are adopted for the MLL datasets including *Ranking loss, Hamming loss, One-error, Coverage* and *Average precision* [38].

---

[2]The detail is provided in Appendix A.4.

Table 2: Predictive performance of each comparing method on MLL datasets in terms of *Ranking loss* (mean ± std). The best performance is highlighted in bold (the smaller the better).

|  | Image | Scene | Yeast | Corel5k | Mirflickr | Delicious |
|---|---|---|---|---|---|---|
| AN | 0.432±0.067 | 0.321±0.113 | 0.383±0.066 | 0.140±0.000 | 0.125±0.002 | 0.131±0.000 |
| AN-LS | 0.378±0.041 | 0.246±0.064 | 0.365±0.031 | 0.186±0.003 | 0.163±0.006 | 0.213±0.007 |
| WAN | 0.354±0.051 | 0.216±0.023 | 0.212±0.021 | 0.129±0.000 | 0.121±0.002 | 0.126±0.000 |
| EPR | 0.401±0.053 | 0.291±0.056 | 0.208±0.010 | 0.139±0.000 | 0.119±0.001 | 0.126±0.000 |
| ROLE | 0.340±0.059 | 0.174±0.028 | 0.213±0.017 | 0.259±0.004 | 0.182±0.014 | 0.336±0.007 |
| EM | 0.471±0.044 | 0.322±0.115 | 0.261±0.030 | 0.155±0.002 | 0.134±0.004 | 0.164±0.001 |
| EM-APL | 0.508±0.028 | 0.420±0.069 | 0.245±0.026 | 0.135±0.001 | 0.138±0.003 | 0.163±0.003 |
| SMILE | 0.260±0.020 | 0.161±0.045 | 0.167±0.002 | 0.125±0.003 | 0.120±0.002 | 0.126±0.000 |
| LL-R | 0.346±0.072 | 0.155±0.021 | 0.227±0.001 | 0.114±0.001 | 0.123±0.003 | 0.129±0.002 |
| LL-CP | 0.329±0.041 | 0.148±0.017 | 0.215±0.000 | 0.114±0.003 | 0.124±0.003 | 0.160±0.001 |
| LL-CT | 0.327±0.019 | 0.180±0.038 | 0.238±0.001 | 0.115±0.001 | 0.124±0.000 | 0.160±0.000 |
| CRISP | **0.164±0.027** | **0.112±0.021** | **0.164±0.001** | **0.113±0.001** | **0.118±0.001** | **0.122±0.000** |

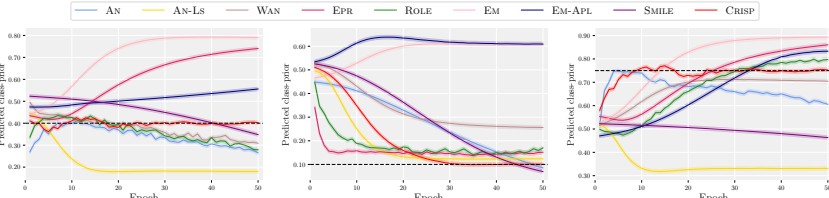

Figure 1: Predicted class-prior of AN[5], AN-LS[5], WAN[5], EPR[5], ROLE[5], EM[5], EM-APL[45], SMILE[38] and CRISP on the 3-*rd* (left), 10-*th* (middle), and 12-*th* labels (right) of the dataset Yeast.

**Comparing methods.** In this paper, CRISP is compared against nine state-of-the-art SPMLL approaches including: 1) AN [5] assumes that the unannotated labels are negative and uses binary cross entropy loss for training. 2) AN-LS [5] assumes that the unannotated labels are negative and reduces the impact of the false negative labels by label smoothing. 3) WAN [5] introduces a weight parameter to down-weight losses in relation to negative labels. 4) EPR [5] utilizes a regularization to constrain the number of predicted positive labels. 5) ROLE [5] online estimates the unannotated labels as learnable parameters throughout training based on EPR with the trick of linear initial. 6) EM [45] reduces the effect of the incorrect labels by the entropy-maximization loss. 7) EM-APL [45] adopts asymmetric-tolerance pseudo-label strategies cooperating with entropy-maximization loss and then more precise supervision can be provided. 8) PLC [33] designs a label-aware global consistency regularization to recover the pseudo-labels leveraging the manifold structure information learned by contrastive learning with data augmentation techniques. 9) SMILE [38] recovers the latent soft labels in a label enhancement process to train the multi-label classifier with binary cross entropy loss. Additionally, since the SPMLL problem is an extreme case of the MLML problem, we employ a state-of-the-art MLML methods as comparative methods: 1) LL [12] treats unobserved labels as noisy labels and dynamically adjusts the threshold to reject or correct samples with a large loss, including three variants LL-R, LL-CT and LL-CP. 2) BOOSTLU [13] apply a BoostLU function to the CAM output of the model to boost the scores of the highlighted regions. It can be integrated with LL. The implementation details are provided in Appendix A.6.

## 5.2 Experimental Results

Table 1 presents the comparison results of CRISP compared with other methods on VOC, COCO, NUS, and CUB. The proposed method achieves the best performance on VOC, COCO, and NUS. Although it does not surpass the top-performing method on CUB, the performance remains competitive. Table 2 record the results of our method and other comparing methods on the MLL datasets in terms of *Ranking loss* respectively. Similar results for other metrics can be found in Appendix A.8. Note that due to the inability to compute the loss function of PLC without data augmentation, we do not report the results of PLC on MLL datasets because data augmentation techniques are not suitable for the MLL datasets. Similarly, since the operations of BOOSTLU for CAM are not applicable to the tabular data in MLL datasets, its results are also not reported. The results demonstrate that our proposed method consistently achieves desirable performance in almost all cases (except the result of Mirflickr on the metric *Average Precision*, where our method attains a comparable performance

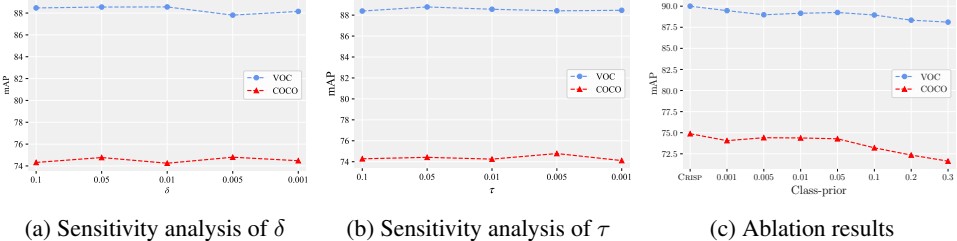

|                    |                    |                    |
|:------------------:|:------------------:|:------------------:|
| (a) Sensitivity analysis of $\delta$ | (b) Sensitivity analysis of $\tau$ | (c) Ablation results |

Figure 2: (a) Parameter sensitivity analysis of $\delta$ (parameter $\tau$ is fixed as $0.01$); (b) Parameter sensitivity analysis of $\tau$ (parameter $\delta$ is fixed as $0.01$); (c) The initial data point represents the performance of the proposed CRISP (with class-priors estimator). The others are the performance with a fixed value for all class-priors gradually increasing from $0.001$ to $0.3$.

against SMILE). Table 12 in Appendix A.10 reports the $p$-values of the wilcoxon signed-ranks test [6] for the corresponding tests and the statistical test results at $0.05$ significance level, which reveals that CRISP consistently outperforms other comparing algorithms (49 out of 55 test cases score **win**). These experimental results validate the effectiveness of CRISP in addressing SPMLL problems.

## 5.3 Further Analysis

### 5.3.1 Class-Prior Prediction

Figure 1 illustrates the comparison results of the predicted class-priors of CRISP with other methods on the 3-*rd* (left), 10-*th* (middle), and 12-*th* labels (right) of the dataset Yeast. Compared with other approaches, whose predicted class-priors $p(\hat{y}_j = 1)$, which represents the expected value of the predicted, significantly deviate from the true class-priors, CRISP achieves consistent predicted class-priors with the ground-truth class-priors (black dashed lines). Without the constraint of the class-priors, the predicted class-prior probability diverges from the true class-prior as epochs increase, significantly impacting the model's performance. In this experiment, the true class-priors are derived by calculating the statistical information for each dataset. More experimental results about the convergence analyses of estimated class-priors of all classes on MLIC datasets are recorded in Appendix A.9. These results demonstrate the necessity of incorporating class-priors in the training of the SPMLL model.

### 5.3.2 Sensitivity Analysis

The performance sensitivity of the proposed CRISP approach with respect to its parameters $\delta$ and $\tau$ during the class-priors estimation phase is analyzed in this section. Figures 2a and 2b illustrate the performance of the proposed method on VOC and COCO under various parameter settings, where $\delta$ and $\tau$ are incremented from $0.001$ to $0.1$. The performance of the proposed method remains consistently stable across a wide range of parameter values. This characteristic is highly desirable as it allows for the robust application of the proposed method without the need for meticulous parameter fine-tuning, ensuring reliable classification results.

### 5.3.3 Ablation Study

Figure 2c depicts the results of the ablation study to investigate the impact of the class-priors estimator by comparing it with a fixed value for all class-priors. The initial data point represents the performance of the proposed CRISP (with class-priors estimator). Subsequently, we maintain a fixed identical class-priors, gradually increasing it from $0.001$ to $0.3$. As expected, our method exhibits superior performance when utilizing the class-priors estimator, compared with employing a fixed class-prior proportion. The ablation results demonstrate the significant enhancement in CRISP performance achieved through the proposed class-priors estimator.

Table 3: Predictive performance comparing CRISP with CRISP-VAL.

| Dataset | CRISP-VAL | CRISP |
|---------|-----------|-------|
| VOC | 89.585±0.318 | **89.820±0.191** |
| COCO | 74.435±0.148 | **74.640±0.219** |
| NUS | 49.230±0.113 | **49.996±0.316** |
| CUB | 19.600±1.400 | **21.650±0.178** |

Table 4: Time cost of class-priors estimation and the whole training time of one epoch.

|  | VOC | COCO | NUS | CUB |
|---|---|---|---|---|
| Time of class-priors estimation (min) | 0.24 | 3.47 | 6.4 | 0.45 |
| Whole training time of one epoch (min) | 2.19 | 27.29 | 49.09 | 3.89 |

Table 5: Predictive performance with different updating frequency of class-priors estimation.

|  | VOC | COCO | NUS | CUB |
|---|---|---|---|---|
| CRISP-3EP | 89.077±0.251 | 73.930±0.399 | 49.463±0.216 | 19.450±0.389 |
| CRISP | 89.820±0.191 | 74.640±0.219 | 49.996±0.316 | 21.650±0.178 |

| Original image | Attention map | | | Original image | Attention map | | |
|---|---|---|---|---|---|---|---|
|  | Observed label | Identified labels | | | Observed label | Identified labels | |
|  | *dog* | *cat* | *sofa* | | *banana* | *apple* | *bottle* |
|  | *cat* | *person* | *bottle* | | *bowl* | *cake* | *fork* |
|  | *bus* | *airplane* | *person* | | *laptop* | *bottle* | *cup* |

Figure 3: Visualization of attention maps on VOC (left) and COCO (right).

Furthermore, we conduct experiments comparing the performance of CRISP with the approach that estimating the class-priors with the full labels of validation set (CRISP-VAL). Table 3 shows that the performance of CRISP is superior to CRISP-VAL. It is indeed feasible to estimate the class-priors using the validation set. However, the size of validation set in many datasets is often quite small, which can lead to unstable estimation of the class-priors, thus leading to a suboptimal performance. Similar results are observed in Table 13 of Appendix A.11 for the MLL datasets.

### 5.3.4 Time Cost of Class-Priors Estimation

In Eq. (2), we have adopted an exhaustive search strategy to find an optimal threshold for estimating class-priors in each training epoch, which may introduce additional computational overhead to the algorithm. We conducted experimental analysis on this aspect. As illustrated in the Table 4, the time for class-priors estimation is short compared to the overall training time for an epoch, ensuring that our method remains practical for use in larger datasets. Additionally, to further enhance the speed of our algorithm, we have experimented with updating the class-priors every few epochs instead of every single one in Table 5. The variant of our method, denoted as CRISP-3EP, updates the priors every three epochs and our experiments show that this results in a negligible loss in performance.

### 5.3.5 Attention Map Visualization

Figure 3 is utilized to visually represent attention maps on COCO, elucidating the underlying mechanism responsible for the efficacy of CRISP in discerning potential positive labels. Specifically, for each original image in the first column, attention maps corresponding to the single observed positive label and identified positive labels are displayed in the subsequent three columns. As evidenced by the figures, given the context of a single positive label, the proposed method demonstrates the ability to identify additional object labels within the image, even for relatively small objects such as the bottle in the first row, the fork in the second row, and the cup in the final row. These observations indicate that the proposed method can accurately detect small objects with the aid of class-priors. This insight further suggests that the proposed method substantially enhances the model's capacity to pinpoint potential positive labels.

## 6 Conclusion

In conclusion, this paper presents a novel approach to address the single-positive multi-label learning (SPMLL) problem by considering the impact of class-priors on the model. We propose a theoretically guaranteed class-priors estimation method that ensures the convergence of estimated class-prior to ground-truth class-priors during the training process. Furthermore, we introduce an unbiased risk estimator based on the estimated class-priors and derive a generalization error bound to guarantee

that the obtained risk minimizer would approximately converge to the optimal risk minimizer of fully supervised learning. Experimental results on ten MLL benchmark datasets demonstrate the effectiveness and superiority of our method over existing SPMLL approaches.

# 7 Acknowledgments

This research was supported by the Jiangsu Science Foundation (BG2024036, BK20243012), the National Science Foundation of China (62576093, 62206050, 62125602, U24A20324, and 92464301), the Fundamental Research Funds for the Central Universities (2242025K30024), and the Big Data Computing Center of Southeast University.

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

# A Supplementary Material

## A.1 Proof of Theorem 4.1

*Proof.* Firstly, we have:

$$
\begin{aligned}
\left| \frac{\hat{q}_j(z)}{\hat{q}_j^p(z)} - \frac{q_j(z)}{q_j^p(z)} \right| &= \frac{\hat{q}_j(z)q_j^p(z) - \hat{q}_j^p(z)q_j(z)}{\hat{q}_j^p(z)q_j^p(z)} \\
&\leq \frac{|\hat{q}_j(z)q_j^p(z) - q_j^p(z)q_j(z)| + |q_j^p(z)q_j(z) - \hat{q}_j^p(z)q_j(z)|}{\hat{q}_j^p(z)q_j^p(z)} \\
&= \frac{1}{\hat{q}_j^p(z)}|\hat{q}_j(z) - q_j(z)| + \frac{q_j(z)}{\hat{q}_j^p(z)q_j^p(z)}|\hat{q}_j^p(z) - q_j^p(z)|,
\end{aligned}
\tag{9}
$$

where $z$ is an arbitrary constant in $[0, 1]$. Using DKW inequality, we have with probability $1 - \delta$: $|\hat{q}_j(z) - q_j(z)| \leq \sqrt{\frac{\log 2/\delta}{2n}}$ and $|\hat{q}_j^p(z) - q_j^p(z)| \leq \sqrt{\frac{\log 2/\delta}{2n_j^p}}$. Therefore, with probability $1 - \delta$:

$$
\left| \frac{\hat{q}_j(z)}{\hat{q}_j^p(z)} - \frac{q_j(z)}{q_j^p(z)} \right| \leq \frac{1}{\hat{q}_j^p(z)} \left( \sqrt{\frac{\log 4/\delta}{2n}} + \frac{q_j(z)}{q_j^p(z)} \sqrt{\frac{\log 4/\delta}{2n_j^p}} \right).
\tag{10}
$$

Then, we define:

$$
\hat{z} = \arg\min_{z \in [0,1]} \left( \frac{\hat{q}_j(z)}{\hat{q}_j^p(z)} + \frac{1 + \tau}{\hat{q}_j^p(z)} \left( \sqrt{\frac{\log(4/\delta)}{2n}} + \sqrt{\frac{\log(4/\delta)}{2n_j^p}} \right) \right),
$$

$$
z^\star = \arg\min_{z \in [0,1]} \frac{q_j(z)}{q_j^p(z)},
$$

$$
\hat{\pi}_j = \frac{\hat{q}_j(\hat{z})}{\hat{q}_j^p(\hat{z})} \qquad \text{and} \qquad \pi_j^\star = \frac{q_j(z^\star)}{q_j^p(z^\star)}.
$$

Next, consider $z' \in [0, 1]$ such that $\hat{q}_j^p(z') = \frac{\tau}{2+\tau}\hat{q}_j^p(z^\star)$. We now show that $\hat{z} < z'$. For any $z \in [0, 1]$, by the DWK inequality, we have with probability $1 - \delta$:

$$
\begin{aligned}
\hat{q}_j^p(z) - \sqrt{\frac{\log 4/\delta}{2n_j^p}} &\leq q_j^p(z), \\
q_j(z) - \sqrt{\frac{\log 4/\delta}{2n}} &\leq \hat{q}_j(z).
\end{aligned}
\tag{11}
$$

Since $\frac{q_j(z^\star)}{q_j^p(z^\star)} \leq \frac{q_j(z)}{q_j^p(z)}$, we have:

$$
\hat{q}_j(z) \geqslant q_j^p(z)\frac{q_j(z^\star)}{q_j^p(z^\star)} - \sqrt{\frac{\log(4/\delta)}{2n}} \geqslant \left( \hat{q}_j^p(z) - \sqrt{\frac{\log(4/\delta)}{2n_j^p}} \right) \frac{q_j(z^\star)}{q_j^p(z^\star)} - \sqrt{\frac{\log(4/\delta)}{2n}}.
\tag{12}
$$

Therefore, we have:

$$
\frac{\hat{q}_j(z)}{\hat{q}_j^p(z)} \geq \pi_j^\star - \frac{1}{\hat{q}_j^p(z)} \left( \sqrt{\frac{\log(4/\delta)}{2n}} + \pi_j^\star \sqrt{\frac{\log(4/\delta)}{2n_j^p}} \right).
\tag{13}
$$

Using Eq. (10) at $z^\star$ and the fact that $\pi_j^\star = \frac{q_j(z^\star)}{q_j^p(z^\star)} \leq 1$, we have:

$$
\begin{aligned}
\frac{\hat{q}_j(z)}{\hat{q}_j^p(z)} &\geq \frac{\hat{q}_j(z^\star)}{\hat{q}_j^p(z^\star)} - \left( \frac{1}{\hat{q}_j^p(z^\star)} + \frac{1}{\hat{q}_j^p(z)} \right) \left( \sqrt{\frac{\log(4/\delta)}{2n}} + \pi_j^\star \sqrt{\frac{\log(4/\delta)}{2n_j^p}} \right) \\
&\geq \frac{\hat{q}_j(z^\star)}{\hat{q}_j^p(z^\star)} - \left( \frac{1}{\hat{q}_j^p(z^\star)} + \frac{1}{\hat{q}_j^p(z)} \right) \left( \sqrt{\frac{\log(4/\delta)}{2n}} + \sqrt{\frac{\log(4/\delta)}{2n_j^p}} \right).
\end{aligned}
\tag{14}
$$

Furthermore, the upper confidence bound at $z$ is lower bounded by:

$$
\frac{\hat{q}_j(z)}{\hat{q}_j^p(z)} + \frac{1+\tau}{\hat{q}_j^p(z)} \left( \sqrt{\frac{\log(4/\delta)}{2n}} + \sqrt{\frac{\log(4/\delta)}{2n_j^p}} \right)
$$

$$
\geq \frac{\hat{q}_j(z^\star)}{\hat{q}_j^p(z^\star)} + \left( \frac{1+\tau}{\hat{q}_j^p(z)} - \frac{1}{\hat{q}_j^p(z^\star)} - \frac{1}{\hat{q}_j^p(z)} \right) \left( \sqrt{\frac{\log(4/\delta)}{2n}} + \sqrt{\frac{\log(4/\delta)}{2n_j^p}} \right) \tag{15}
$$

$$
= \frac{\hat{q}_j(z^\star)}{\hat{q}_j^p(z^\star)} + \left( \frac{\tau}{\hat{q}_j^p(z)} - \frac{1}{\hat{q}_j^p(z^\star)} \right) \left( \sqrt{\frac{\log(4/\delta)}{2n}} + \sqrt{\frac{\log(4/\delta)}{2n_j^p}} \right).
$$

Using Eq. (15) at $z = z'$ where $\hat{q}_j^p(z') = \frac{\tau}{2+\tau}\hat{q}_j^p(z^\star)$, we have:

$$
\frac{\hat{q}_j(z')}{\hat{q}_j^p(z')} + \frac{1+\tau}{\hat{q}_j^p(z')} \left( \sqrt{\frac{\log(4/\delta)}{2n}} + \sqrt{\frac{\log(4/\delta)}{2n_j^p}} \right)
$$

$$
\geq \frac{\hat{q}_j(z^\star)}{\hat{q}_j^p(z^\star)} + \left( \frac{\tau}{\hat{q}_j^p(z')} - \frac{1}{\hat{q}_j^p(z^\star)} \right) \left( \sqrt{\frac{\log(4/\delta)}{2n}} + \sqrt{\frac{\log(4/\delta)}{2n_j^p}} \right). \tag{16}
$$

$$
\geq \frac{\hat{q}_j(z^\star)}{\hat{q}_j^p(z^\star)} + \frac{1+\tau}{\hat{q}_j^p(z^\star)} \left( \sqrt{\frac{\log(4/\delta)}{2n}} + \sqrt{\frac{\log(4/\delta)}{2n_j^p}} \right).
$$

Moreover from Eq. (15) and using definition of $\hat{z}$, we have:

$$
\frac{\hat{q}_j(z')}{\hat{q}_j^p(z')} + \frac{1+\tau}{\hat{q}_j^p(z')} \left( \sqrt{\frac{\log(4/\delta)}{2n}} + \sqrt{\frac{\log(4/\delta)}{2n_j^p}} \right)
$$

$$
\geq \frac{\hat{q}_j(z^\star)}{\hat{q}_j^p(z^\star)} + \frac{1+\tau}{\hat{q}_j^p(z^\star)} \left( \sqrt{\frac{\log(4/\delta)}{2n}} + \sqrt{\frac{\log(4/\delta)}{2n_j^p}} \right) \tag{17}
$$

$$
\geq \frac{\hat{q}_j(\hat{z})}{\hat{q}_j^p(\hat{z})} + \frac{1+\tau}{\hat{q}_j^p(\hat{z})} \left( \sqrt{\frac{\log(4/\delta)}{2n}} + \sqrt{\frac{\log(4/\delta)}{2n_j^p}} \right),
$$

and hence $\hat{z} \leq z'$.

We now establish an upper and lower bound on $\hat{z}$. By definition of $\hat{z}$, we have:

$$
\frac{\hat{q}_j(\hat{z})}{\hat{q}_j^p(\hat{z})} + \frac{1+\tau}{\hat{q}_j^p(\hat{z})} \left( \sqrt{\frac{\log(4/\delta)}{2n}} + \sqrt{\frac{\log(4/\delta)}{2n_j^p}} \right)
$$

$$
\leq \min_{z \in [0,1]} \left( \frac{\hat{q}_j(z)}{\hat{q}_j^p(z)} + \frac{1+\tau}{\hat{q}_j^p(z)} \left( \sqrt{\frac{\log(4/\delta)}{2n}} + \sqrt{\frac{\log(4/\delta)}{2n_j^p}} \right) \right) \tag{18}
$$

$$
\leq \frac{\hat{q}_j(z^\star)}{\hat{q}_j^p(z^\star)} + \frac{1+\tau}{\hat{q}_j^p(z^\star)} \left( \sqrt{\frac{\log(4/\delta)}{2n}} + \sqrt{\frac{\log(4/\delta)}{2n_j^p}} \right).
$$

Using Eq. (10) at $z^\star$, we have:

$$
\frac{\hat{q}_j(z^\star)}{\hat{q}_j^p(z^\star)} \leq \frac{q_j(z^\star)}{q_j^p(z^\star)} + \frac{1}{\hat{q}_j^p(z^\star)} \left( \sqrt{\frac{\log(4/\delta)}{2n}} + \pi_j^\star \sqrt{\frac{\log(4/\delta)}{2n_j^p}} \right). \tag{19}
$$

Then, we have:

$$
\hat{\pi}_j = \frac{\hat{q}_j(\hat{z})}{\hat{q}_j^p(\hat{z})} \leq \pi_j^\star + \frac{2+\tau}{\hat{q}_j^p(z^\star)} \left( \sqrt{\frac{\log(4/\delta)}{2n}} + \sqrt{\frac{\log(4/\delta)}{2n_j^p}} \right). \tag{20}
$$

Assume $n_j^p \geq 2\frac{\log 4/\delta}{q_j^{p\,2}(z^\star)}$, we have $\hat{q}_j^p(z^\star) \geq q_j^p(z^\star)/2$ and hence:

$$
\hat{\pi}_j \leq \pi_j^\star + \frac{4+2\tau}{q_j^p(z^\star)} \left( \sqrt{\frac{\log(4/\delta)}{2n}} + \sqrt{\frac{\log(4/\delta)}{2n_j^p}} \right). \tag{21}
$$

From Eq. (10) at $\hat{z}$, we have:

$$\frac{q_j(\hat{z})}{q_j^p(\hat{z})} \leq \frac{\hat{q}_j(\hat{z})}{\hat{q}_j^p(\hat{z})} + \frac{1}{\hat{q}_j^p(\hat{z})}\left(\sqrt{\frac{\log(4/\delta)}{2n}} + \frac{q_j(\hat{z})}{q_j^p(\hat{z})}\sqrt{\frac{\log(4/\delta)}{2n_j^p}}\right). \tag{22}$$

Since $\pi_j^\star \leq \frac{q_j(\hat{z})}{q_j^p(\hat{z})}$, we have:

$$\pi_j^\star \leq \frac{q_j(\hat{z})}{q_j^p(\hat{z})} \leq \frac{\hat{q}_j(\hat{z})}{\hat{q}_j^p(\hat{z})} + \frac{1}{\hat{q}_j^p(\hat{z})}\left(\sqrt{\frac{\log(4/\delta)}{2n}} + \frac{q_j(\hat{z})}{q_j^p(\hat{z})}\sqrt{\frac{\log(4/\delta)}{2n_j^p}}\right). \tag{23}$$

Using Eq. (21) and the assumption that $n \geq n_j^p \geq 2\frac{\log 4/\delta}{q_j^{p2}(z^\star)}$ , we have:

$$\begin{aligned}
\hat{\pi}_j = \frac{\hat{q}_j(\hat{z})}{\hat{q}_j^p(\hat{z})} &\leq \pi_j^\star + \frac{4 + 2\tau}{q_j^p(z^\star)}\left(\sqrt{\frac{\log(4/\delta)}{2n}} + \sqrt{\frac{\log(4/\delta)}{2n_j^p}}\right) \\
&\leq \pi_j^\star + 4 + 2\tau \leq 1 + 4 + 2\tau = 5 + 2\tau.
\end{aligned} \tag{24}$$

Using this in Eq. (23), we have:

$$\pi_j^\star \leq \frac{\hat{q}_j(\hat{z})}{\hat{q}_j^p(\hat{z})} + \frac{1}{\hat{q}_j^p(\hat{z})}\left(\sqrt{\frac{\log(4/\delta)}{2n}} + (5 + 2\tau)\sqrt{\frac{\log(4/\delta)}{2n_j^p}}\right). \tag{25}$$

Since $\hat{z} \leq z'$, we have $\hat{q}_j^p(\hat{z}) \geq \hat{q}_j^p(z') = \frac{\tau}{2+\tau}\hat{q}_j^p(z^\star)$. Therefore, we have:

$$\pi_j^\star - \frac{2 + \tau}{\tau\hat{q}_j^p(z^\star)}\left(\sqrt{\frac{\log(4/\delta)}{2n}} + (5 + 2\tau)\sqrt{\frac{\log(4/\delta)}{2n_j^p}}\right) \leq \frac{\hat{q}_j(\hat{z})}{\hat{q}_j^p(\hat{z})} = \hat{\pi}_j. \tag{26}$$

With the assumption that $n_j^p \geq 2\frac{\log 4/\delta}{q_j^{p2}(z^\star)}$, we have $\hat{q}_j^p(z^\star) \geq q_j^p(z^\star)/2$, which implies:

$$\pi_j^\star - \frac{4 + 2\tau}{\tau q_j^p(z^\star)}\left(\sqrt{\frac{\log(4/\delta)}{2n}} + (5 + 2\tau)\sqrt{\frac{\log(4/\delta)}{2n_j^p}}\right) \leq \hat{\pi}_j. \tag{27}$$

Note that since $\pi_j \leq \pi_j^\star$, the lower bound remains the same as in Theorem 4.1. For the upper bound, with $q_j(z^\star) = \pi_j q_j^p(z^\star) + (1 - \pi_j)q_j^n(z^\star)$, we have $\pi_j^\star = \pi_j + (1 - \pi_j)\frac{q_j^n(z^\star)}{q_j^p(z^\star)}$. Then the proof is completed.

$\square$

## A.2 The details of the optimization of Eq. (2)

In practice, to determine the optimal threshold, we conduct an exhaustive search across the set of outputs generated by the function $f^j$ for each class. For instance, for a given class $j$, and a set of instances $\boldsymbol{x}_1, \boldsymbol{x}_2, \boldsymbol{x}_3$ in our dataset, we compute the corresponding outputs $z_1 = f^j(\boldsymbol{x}_1), z_2 = f^j(\boldsymbol{x}_2), z_3 = f^j(\boldsymbol{x}_3)$.

The optimal threshold $\hat{z}$ is then selected by identifying the value of $z \in \{z_1, z_2, z_3\}$ that minimizes the objective function specified in Equation (2):

$$\hat{z} = \arg\min_{z \in \{z_1, z_2, z_3\}}\left(\frac{\hat{q}_j(z)}{\hat{q}_j^p(z)} + \frac{1 + \tau}{\hat{q}_j^p(z)}\left(\sqrt{\frac{\log(4/\delta)}{2n}} + \sqrt{\frac{\log(4/\delta)}{2n_j^p}}\right)\right)$$

This approach ensures that we find the optimal threshold that minimizes the given expression, as per Eq. (2), across all available output values from the function $f^j$.

### A.3 Details of Eq. (3)

$$
\begin{aligned}
\mathcal{R}(f) &= \mathbb{E}_{(\boldsymbol{x},\boldsymbol{y})\sim p(\boldsymbol{x},\boldsymbol{y})}\left[\mathcal{L}(f(\boldsymbol{x}),\boldsymbol{y})\right] \\
&= \int_{\boldsymbol{x}}\sum_{\boldsymbol{y}}\mathcal{L}(f(\boldsymbol{x}),\boldsymbol{y})p(\boldsymbol{x}|\boldsymbol{y})p(\boldsymbol{y})d\boldsymbol{x} \\
&= \sum_{\boldsymbol{y}}p(\boldsymbol{y})\int_{\boldsymbol{x}}\mathcal{L}(f(\boldsymbol{x}),\boldsymbol{y})p(\boldsymbol{x}|\boldsymbol{y})d\boldsymbol{x} \\
&= \sum_{\boldsymbol{y}}p(\boldsymbol{y})\mathbb{E}_{\boldsymbol{x}\sim p(\boldsymbol{x}|\boldsymbol{y})}\left[\mathcal{L}(f(\boldsymbol{x}),\boldsymbol{y})\right].
\end{aligned}
\tag{28}
$$

### A.4 Details of Eq. (5)

The absolute loss function is $\ell(f^j(\boldsymbol{x}),y_j) = |f^j(\boldsymbol{x}) - y_j|$, when $y_j = 1$, $\ell(f^j(\boldsymbol{x}),1) = |1 - f^j(\boldsymbol{x})|$, and when $y_j = 0$, $\ell(f^j(\boldsymbol{x}),0) = f^j(\boldsymbol{x})$. Then:

$$
\begin{aligned}
\mathcal{R}(f) &= \sum_{\boldsymbol{y}}p(\boldsymbol{y})\mathbb{E}_{\boldsymbol{x}\sim p(\boldsymbol{x}|\boldsymbol{y})}\left[\sum_{j=1}^{c}y_j\ell(f^j(\boldsymbol{x}),1) + (1-y_j)\ell(f^j(\boldsymbol{x}),0)\right] \\
&= \sum_{j=1}^{c}p(y_j=1)\mathbb{E}_{\boldsymbol{x}\sim p(\boldsymbol{x}|y_j=1)}\left[\ell(f^j(\boldsymbol{x}),1)\right] + p(y_j=0)\mathbb{E}_{\boldsymbol{x}\sim p(\boldsymbol{x}|y_j=0)}\left[\ell(f^j(\boldsymbol{x}),0)\right] \\
&= \sum_{j=1}^{c}p(y_j=1)\mathbb{E}_{\boldsymbol{x}\sim p(\boldsymbol{x}|y_j=1)}\left[1 - f^j(\boldsymbol{x})\right] + (1 - p(y_j=1))\mathbb{E}_{\boldsymbol{x}\sim p(\boldsymbol{x}|y_j=0)}\left[f^j(\boldsymbol{x})\right] \\
&= \sum_{j=1}^{c}p(y_j=1)\mathbb{E}_{\boldsymbol{x}\sim p(\boldsymbol{x}|y_j=1)}\left[1 - f^j(\boldsymbol{x})\right] + \mathbb{E}_{\boldsymbol{x}\sim p(\boldsymbol{x})}\left[f^j(\boldsymbol{x})\right] \\
&\quad - p(y_j=1)\mathbb{E}_{\boldsymbol{x}\sim p(\boldsymbol{x}|y_j=1)}\left[f^j(\boldsymbol{x})\right] \\
&= \sum_{j=1}^{c}p(y_j=1)\mathbb{E}_{\boldsymbol{x}\sim p(\boldsymbol{x}|y_j=1)}\left[1 - f^j(\boldsymbol{x})\right] + \mathbb{E}_{\boldsymbol{x}\sim p(\boldsymbol{x})}\left[f^j(\boldsymbol{x})\right] \\
&\quad - p(y_j=1)\mathbb{E}_{\boldsymbol{x}\sim p(\boldsymbol{x}|y_j=1)}\left[f^j(\boldsymbol{x}) - 1 + 1\right] \\
&= \sum_{j=1}^{c}2p(y_j=1)\mathbb{E}_{\boldsymbol{x}\sim p(\boldsymbol{x}|y_j=1)}\left[1 - f^j(\boldsymbol{x})\right] + \mathbb{E}_{\boldsymbol{x}\sim p(\boldsymbol{x})}\left[f^j(\boldsymbol{x})\right] - p(y_j=1).
\end{aligned}
\tag{29}
$$

### A.5 Proof of Theorem 4.2

In this subsection, an estimation error bound is established for Eq. (7) to demonstrate its learning consistency. Specifically, The derivation of the estimation error bound involves two main parts, each corresponding to one of the loss terms in Eq. (7). The empirical risk estimator according to Eq. (7) can be written as:

$$
\begin{aligned}
\widehat{\mathcal{R}}_{sp}(f) &= \sum_{j=1}^{c}\frac{2\pi_j}{|\mathcal{S}_{L_j}|}\sum_{\boldsymbol{x}\in\mathcal{S}_{L_j}}\left(1 - f^j(\boldsymbol{x})\right) + \frac{1}{n}\sum_{\boldsymbol{x}\in\tilde{\mathcal{D}}}\left(f^j(\boldsymbol{x}) - \pi_j\right) \\
&= \widehat{\mathcal{R}}_{sp}^{L}(f) + \widehat{\mathcal{R}}_{sp}^{U}(f),
\end{aligned}
\tag{30}
$$

Firstly, we define the function spaces as:

$$
\mathcal{G}_{sp}^{L} = \left\{(\boldsymbol{x},\boldsymbol{l})\mapsto\sum_{j=1}^{c}2\pi_j l_j\left(1 - f^j(\boldsymbol{x})\right)|f\in\mathcal{F}\right\}, \mathcal{G}_{sp}^{U} = \left\{(\boldsymbol{x},\boldsymbol{l})\mapsto\sum_{j=1}^{c}\left(f^j(\boldsymbol{x}) - \pi_j\right)|f\in\mathcal{F}\right\},
$$

and denote the expected Rademacher complexity [26] of the function spaces as:

$$\widetilde{\mathfrak{R}}_n \left(\mathcal{G}_{sp}^L\right) = \mathbb{E}_{\boldsymbol{x},\boldsymbol{l},\boldsymbol{\sigma}} \left[\sup_{g \in \mathcal{G}_{sp}^L} \sum_{i=1}^n \sigma_i g\left(\boldsymbol{x}_i, \boldsymbol{l}_i\right)\right],$$

$$\widetilde{\mathfrak{R}}_n \left(\mathcal{G}_{sp}^U\right) = \mathbb{E}_{\boldsymbol{x},\boldsymbol{l},\boldsymbol{\sigma}} \left[\sup_{g \in \mathcal{G}_{sp}^U} \sum_{i=1}^n \sigma_i g\left(\boldsymbol{x}_i, \boldsymbol{l}_i\right)\right],$$

where $\boldsymbol{\sigma} = \{\sigma_1, \sigma_2, \cdots, \sigma_n\}$ is $n$ Rademacher variables with $\sigma_i$ independently uniform variable taking value in $\{+1, -1\}$. Then we have:

**Lemma A.1.** *We suppose that the loss function* $\mathcal{L}_{sp}^L = \sum_{j=1}^c 2\pi_j l_j \left(1 - f^j(\boldsymbol{x})\right)$ *and* $\mathcal{L}_{sp}^U = \sum_{j=1}^c \left(f^j(\boldsymbol{x}) - \pi_j\right)$ *could be bounded by* $M$, *i.e.,* $M = \sup_{\boldsymbol{x} \in \mathcal{X}, f \in \mathcal{F}, \boldsymbol{l} \in \mathcal{Y}} \max(\mathcal{L}_{sp}^L(f(\boldsymbol{x}), \boldsymbol{l}), \mathcal{L}_{sp}^U(f(\boldsymbol{x}), \boldsymbol{l}))$, *and for any* $\delta > 0$, *with probability at least* $1 - \delta$, *we have:*

$$\sup_{f \in \mathcal{F}} |\mathcal{R}_{sp}^L(f) - \hat{\mathcal{R}}_{sp}^L(f)| \leq \frac{2}{C} \widetilde{\mathfrak{R}}_n \left(\mathcal{G}_{sp}^L\right) + \frac{M}{2 \min_j |\mathcal{S}_{L_j}|} \sqrt{\frac{\log \frac{2}{\delta}}{2n}},$$

$$\sup_{f \in \mathcal{F}} |\mathcal{R}_{sp}^U(f) - \hat{\mathcal{R}}_{sp}^U(f)| \leq 2\widetilde{\mathfrak{R}}_n \left(\mathcal{G}_{sp}^U\right) + \frac{M}{2} \sqrt{\frac{\log \frac{2}{\delta}}{2n}},$$

where $\mathcal{R}_{sp}^L(f) = \sum_{j=1}^c 2\pi_j \mathbb{E}_{\boldsymbol{x} \sim p(\boldsymbol{x}|y_j=1)} \left[1 - f^j(\boldsymbol{x})\right]$, $\mathcal{R}_{sp}^U(f) = \mathbb{E}_{\boldsymbol{x} \sim p(\boldsymbol{x})} \sum_{j=1}^c \left[f^j(\boldsymbol{x})\right] - \pi_j$ and $C = \min_j \mathbb{E}_{\tilde{\mathcal{D}}} \left[\sum_{i=1}^n l_i^j\right]$ is a constant.

*Proof.* Suppose an example $(\boldsymbol{x}, \boldsymbol{l})$ is replaced by another arbitrary example $(\boldsymbol{x}', \boldsymbol{l}')$, then the change of $\sup_{f \in \mathcal{F}} \mathcal{R}_{sp}^L(f) - \hat{\mathcal{R}}_{sp}^L(f)$ is no greater than $\frac{M}{2n \min_j |\mathcal{S}_{L_j}|}$. By applying McDiarmid's inequality, for any $\delta > 0$, with probility at least $1 - \frac{\delta}{2}$,

$$\sup_{f \in \mathcal{F}} \mathcal{R}_{sp}^L(f) - \hat{\mathcal{R}}_{sp}^L(f) \leq \mathbb{E} \left[\sup_{f \in \mathcal{F}} \mathcal{R}_{sp}^L(f) - \hat{\mathcal{R}}_{sp}^L(f)\right] + \frac{M}{2 \min_j |\mathcal{S}_{L_j}|} \sqrt{\frac{\log \frac{2}{\delta}}{2n}}.$$

By symmetry, we can obtain

$$\sup_{f \in \mathcal{F}} |\mathcal{R}_{sp}^L(f) - \hat{\mathcal{R}}_{sp}^L(f)| \leq \mathbb{E} \left[\sup_{f \in \mathcal{F}} \mathcal{R}_{sp}^L(f) - \hat{\mathcal{R}}_{sp}^L(f)\right] + \frac{M}{2 \min_j |\mathcal{S}_{L_j}|} \sqrt{\frac{\log \frac{2}{\delta}}{2n}}.$$

Next is to bound the term $\mathbb{E}\left[\sup_{f\in\mathcal{F}}\mathcal{R}_{sp}^L(f)-\hat{\mathcal{R}}_{sp}^L(f)\right]$:

$$\mathbb{E}\left[\sup_{f\in\mathcal{F}}\mathcal{R}_{sp}^L(f)-\hat{\mathcal{R}}_{sp}^L(f)\right]=\mathbb{E}_{\tilde{\mathcal{D}}}\left[\sup_{f\in\mathcal{F}}\mathcal{R}_{sp}^L(f)-\hat{\mathcal{R}}_{sp}^L(f)\right]$$

$$=\mathbb{E}_{\tilde{\mathcal{D}}}\left[\sup_{f\in\mathcal{F}}\mathbb{E}_{\tilde{\mathcal{D}}'}\left[\hat{\mathcal{R}}_{sp}'^L(f)-\hat{\mathcal{R}}_{sp}^L(f)\right]\right]$$

$$\leq\mathbb{E}_{\tilde{\mathcal{D}},\tilde{\mathcal{D}}'}\left[\sup_{f\in\mathcal{F}}\left[\hat{\mathcal{R}}_{sp}'^L(f)-\hat{\mathcal{R}}_{sp}^L(f)\right]\right]$$

$$=\mathbb{E}_{\tilde{\mathcal{D}},\tilde{\mathcal{D}}',\boldsymbol{\sigma}}\left[\sup_{f\in\mathcal{F}}\sum_{i=1}^{n}\sum_{j=1}^{c}\sigma_i\left(\frac{2\pi_j}{\sum_{i=1}^{n}l_i'^j}l_i'^j\left(1-f^j(\boldsymbol{x}_i')\right)-\frac{2\pi_j}{\sum_{i=1}^{n}l_i^j}l_i^j\left(1-f^j(\boldsymbol{x}_i)\right)\right)\right]$$

$$\leq\mathbb{E}_{\tilde{\mathcal{D}}',\boldsymbol{\sigma}}\left[\sup_{f\in\mathcal{F}}\sum_{i=1}^{n}\sum_{j=1}^{c}\sigma_i\left(\frac{2\pi_j}{\sum_{i=1}^{n}l_i'^j}l_i'^j\left(1-f^j(\boldsymbol{x}_i')\right)\right)\right]$$

$$+\mathbb{E}_{\tilde{\mathcal{D}},\boldsymbol{\sigma}}\left[\sup_{f\in\mathcal{F}}\sum_{i=1}^{n}\sum_{j=1}^{c}\sigma_i\left(\frac{2\pi_j}{\sum_{i=1}^{n}l_i^j}l_i^j\left(1-f^j(\boldsymbol{x}_i)\right)\right)\right]$$

$$\leq\frac{1}{C}\mathbb{E}_{\tilde{\mathcal{D}}',\boldsymbol{\sigma}}\left[\sup_{f\in\mathcal{F}}\sum_{i=1}^{n}\sum_{j=1}^{c}\sigma_i\left(2\pi_j l_i'^j\left(1-f^j(\boldsymbol{x}_i')\right)\right)\right]$$

$$+\frac{1}{C}\mathbb{E}_{\tilde{\mathcal{D}},\boldsymbol{\sigma}}\left[\sup_{f\in\mathcal{F}}\sum_{i=1}^{n}\sum_{j=1}^{c}\sigma_i\left(2\pi_j l_i^j\left(1-f^j(\boldsymbol{x}_i)\right)\right)\right]$$

$$=\frac{2}{C}\widetilde{\mathfrak{R}}_n\left(\mathcal{G}_{sp}^L\right),$$

where $C$ is a constant that $C=\min_j\mathbb{E}_{\tilde{\mathcal{D}}}\left[\sum_{i=1}^{n}y_i^j\right]$. Then we have:

$$\sup_{f\in\mathcal{F}}|\mathcal{R}_{sp}^L(f)-\hat{\mathcal{R}}_{sp}^L(f)|\leq\frac{2}{C}\widetilde{\mathfrak{R}}_n\left(\mathcal{G}_{sp}^L\right)+\frac{M}{2\min_j|\mathcal{S}_{L_j}|}\sqrt{\frac{\log\frac{2}{\delta}}{2n}}.$$

Similarly, we can obtain:

$$\sup_{f\in\mathcal{F}}|\mathcal{R}_{sp}^U(f)-\hat{\mathcal{R}}_{sp}^U(f)|\leq 2\widetilde{\mathfrak{R}}_n\left(\mathcal{G}_{sp}^U\right)+\frac{M}{2}\sqrt{\frac{\log\frac{2}{\delta}}{2n}},$$

$\square$

**Lemma A.2.** *Define* $\rho=\max_j 2\pi_j$, $\mathcal{H}_j=\left\{h:\boldsymbol{x}\mapsto f^j(\boldsymbol{x})|f\in\mathcal{F}\right\}$ *and* $\mathfrak{R}_n\left(\mathcal{H}_j\right)=\mathbb{E}_{p(\boldsymbol{x})}\mathbb{E}_{\boldsymbol{\sigma}}\left[\sup_{h\in\mathcal{H}_j}\frac{1}{n}\sum_{i=1}^{n}h\left(\boldsymbol{x}_i\right)\right]$. *Then, we have with Rademacher vector contraction inequality:*

$$\widetilde{\mathfrak{R}}_n\left(\mathcal{G}_{sp}^L\right)\leq\sqrt{2}\rho\sum_{j=1}^{c}\mathfrak{R}_n(\mathcal{H}_j),\qquad\widetilde{\mathfrak{R}}_n\left(\mathcal{G}_{sp}^U\right)\leq\sqrt{2}\sum_{j=1}^{c}\mathfrak{R}_n(\mathcal{H}_j),$$

Based on Lemma A.1 and Lemma A.2, we could obtain the following theorem.

**Theorem A.3.** *Assume the loss function* $\mathcal{L}_{sp}^L=\sum_{j=1}^{c}2\pi_j l_j\left(1-f^j(\boldsymbol{x})\right)$ *and* $\mathcal{L}_{sp}^U=\sum_{j=1}^{c}\left(f^j(\boldsymbol{x})-\pi_j\right)$ *could be bounded by* $M$, *i.e.,* $M=$

Table 6: Characteristics of the MLIC datasets.

| Dataset | #Training | #Validation | #Testing | #Classes |
|---------|-----------|-------------|----------|----------|
| VOC | 4574 | 1143 | 5823 | 20 |
| COCO | 65665 | 16416 | 40137 | 80 |
| NUS | 120000 | 30000 | 60260 | 81 |
| CUB | 4795 | 1199 | 5794 | 312 |

Table 7: Characteristics of the MLL datasets.

| Dataset | #Examples | #Features | #Classes | #Domain |
|---------|-----------|-----------|----------|---------|
| Image | 2000 | 294 | 5 | Images |
| Scene | 2407 | 294 | 6 | Images |
| Yeast | 2417 | 103 | 14 | Biology |
| Corel5k | 5000 | 499 | 374 | Images |
| Mirflickr | 24581 | 1000 | 38 | Images |
| Delicious | 16091 | 500 | 983 | Text |

$\sup_{\boldsymbol{x} \in \mathcal{X}, f \in \mathcal{F}, \boldsymbol{l} \in \mathcal{Y}} \max(\mathcal{L}_{sp}^{L}(f(\boldsymbol{x}), \boldsymbol{l}), \mathcal{L}_{sp}^{U}(f(\boldsymbol{x}), \boldsymbol{y}))$, *with probability at least* $1 - \delta$, *we have:*

$$\mathcal{R}(\hat{f}_{sp}) - \mathcal{R}(f^{\star}) \leq \frac{4}{C} \sum_{j=1}^{c} \widetilde{\mathfrak{R}}_n \left( \mathcal{G}_{sp}^{L} \right) + \frac{M}{\min_j |\mathcal{S}_{L_j}|} \sqrt{\frac{\log \frac{4}{\delta}}{2n}} + 4\widetilde{\mathfrak{R}}_n \left( \mathcal{G}_{sp}^{U} \right) + M \sqrt{\frac{\log \frac{4}{\delta}}{2n}}$$

$$\leq \frac{4\sqrt{2}\rho}{C} \sum_{j=1}^{c} \mathfrak{R}_n(\mathcal{H}_j) + \frac{M}{\min_j |\mathcal{S}_{L_j}|} \sqrt{\frac{\log \frac{4}{\delta}}{2n}} + 4\sqrt{2} \sum_{j=1}^{c} \mathfrak{R}_n(\mathcal{H}_j) + M \sqrt{\frac{\log \frac{4}{\delta}}{2n}}.$$

*Proof.*

$$\mathcal{R}(\hat{f}_{sp}) - \mathcal{R}(f^{\star}) = \mathcal{R}(\hat{f}_{sp}) - \hat{\mathcal{R}}_{sp}(\hat{f}) + \hat{\mathcal{R}}_{sp}(\hat{f}) - \hat{\mathcal{R}}_{sp}(f^{\star}) + \hat{\mathcal{R}}_{sp}(f^{\star}) - \mathcal{R}(f^{\star})$$

$$\leq \mathcal{R}(\hat{f}_{sp}) - \hat{\mathcal{R}}_{sp}(\hat{f}) + \hat{\mathcal{R}}_{sp}(f^{\star}) - \mathcal{R}(f^{\star})$$

$$= \mathcal{R}_{sp}^{L}(\hat{f}_{sp}) - \hat{\mathcal{R}}_{sp}^{L}(\hat{f}) + \hat{\mathcal{R}}_{sp}^{L}(f^{\star}) - \mathcal{R}_{sp}^{L}(f^{\star})$$

$$+ \mathcal{R}_{sp}^{U}(\hat{f}_{sp}) - \hat{\mathcal{R}}_{sp}^{U}(\hat{f}) + \hat{\mathcal{R}}_{sp}^{U}(f^{\star}) - \mathcal{R}_{sp}^{U}(f^{\star})$$

$$\leq 2 \sup_{f \in \mathcal{F}} |\mathcal{R}_{sp}^{L}(f) - \hat{\mathcal{R}}_{sp}^{L}(f)| + 2 \sup_{f \in \mathcal{F}} |\mathcal{R}_{sp}^{U}(f) - \hat{\mathcal{R}}_{sp}^{U}(f)|$$

$$\leq \frac{4}{C} \widetilde{\mathfrak{R}}_n \left( \mathcal{G}_{sp}^{L} \right) + \frac{M}{\min_j |\mathcal{S}_{L_j}|} \sqrt{\frac{\log \frac{4}{\delta}}{2n}} + 4\widetilde{\mathfrak{R}}_n \left( \mathcal{G}_{sp}^{U} \right) + M \sqrt{\frac{\log \frac{4}{\delta}}{2n}}$$

$$\leq \frac{4\sqrt{2}\rho}{C} \sum_{j=1}^{c} \mathfrak{R}_n(\mathcal{H}_j) + \frac{M}{\min_j |\mathcal{S}_{L_j}|} \sqrt{\frac{\log \frac{4}{\delta}}{2n}} + 4\sqrt{2} \sum_{j=1}^{c} \mathfrak{R}_n(\mathcal{H}_j) + M \sqrt{\frac{\log \frac{4}{\delta}}{2n}}.$$

$\square$

## A.6 Implementation Details

During the implementation, we first initialize the predictive network by performing warm-up training with AN solution, which could facilitate learning a fine network in the early stages. Furthermore, after each epoch, the class prior is reestimated via the trained model. The code implementation is based on PyTorch, and the experiments are conducted on GeForce RTX 3090 GPUs. The batch size is selected from $\{8, 16\}$ and the number of epochs is set to 10. The learning rate and weight decay are selected from $\{10^{-2}, 10^{-3}, 10^{-4}, 10^{-5}\}$ with a validation set. The hyperparameters $\delta$ and $\tau$ are all fixed as 0.01. All the comparing methods run 5 trials on each datasets. For fairness, we employed ResNet-50 as the backbone for all comparing methods.

Table 8: Predictive performance of each comparing method on MLL datasets in terms of *Average Precision* (mean ± std). The best performance is highlighted in bold (the larger the better).

| | Image | Scene | Yeast | Corel5k | Mirflickr | Delicious |
|---|---|---|---|---|---|---|
| AN | 0.534±0.061 | 0.580±0.104 | 0.531±0.079 | 0.217±0.003 | 0.615±0.004 | 0.317±0.002 |
| AN-LS | 0.574±0.037 | 0.631±0.072 | 0.538±0.044 | 0.230±0.002 | 0.587±0.006 | 0.261±0.006 |
| WAN | 0.576±0.041 | 0.661±0.033 | 0.698±0.017 | 0.241±0.002 | 0.621±0.004 | 0.315±0.000 |
| EPR | 0.539±0.028 | 0.597±0.062 | 0.710±0.008 | 0.214±0.001 | 0.628±0.003 | 0.314±0.000 |
| ROLE | 0.606±0.041 | 0.700±0.040 | 0.711±0.013 | 0.203±0.003 | 0.516±0.027 | 0.130±0.003 |
| EM | 0.486±0.031 | 0.549±0.103 | 0.642±0.029 | 0.294±0.002 | 0.614±0.003 | 0.293±0.001 |
| EM-APL | 0.467±0.026 | 0.448±0.049 | 0.654±0.040 | 0.275±0.003 | 0.589±0.007 | 0.311±0.001 |
| SMILE | 0.670±0.021 | 0.722±0.071 | 0.751±0.004 | 0.295±0.004 | **0.629±0.003** | 0.318±0.001 |
| LL-R | 0.605±0.058 | 0.714±0.035 | 0.658±0.006 | 0.268±0.002 | 0.625±0.001 | 0.296±0.004 |
| LL-CP | 0.595±0.031 | 0.735±0.028 | 0.700±0.000 | 0.259±0.004 | 0.621±0.007 | 0.251±0.007 |
| LL-CT | 0.600±0.012 | 0.669±0.052 | 0.629±0.007 | 0.258±0.004 | 0.619±0.004 | 0.253±0.004 |
| CRISP | **0.749±0.037** | **0.795±0.031** | **0.758±0.002** | **0.304±0.003** | 0.628±0.003 | **0.319±0.001** |

Table 9: Predictive performance of each comparing method on MLL datasets in terms of *Coverage* (mean ± std). The best performance is highlighted in bold (the smaller the better).

| | Image | Scene | Yeast | Corel5k | Mirflickr | Delicious |
|---|---|---|---|---|---|---|
| AN | 0.374±0.050 | 0.279±0.094 | 0.707±0.045 | 0.330±0.001 | 0.342±0.003 | 0.653±0.001 |
| AN-LS | 0.334±0.033 | 0.217±0.052 | 0.703±0.012 | 0.441±0.009 | 0.433±0.015 | 0.830±0.016 |
| WAN | 0.313±0.040 | 0.192±0.019 | 0.512±0.045 | 0.309±0.001 | 0.334±0.002 | 0.632±0.001 |
| EPR | 0.352±0.043 | 0.254±0.046 | 0.506±0.011 | 0.328±0.001 | 0.332±0.002 | 0.637±0.001 |
| ROLE | 0.306±0.049 | 0.157±0.023 | 0.519±0.026 | 0.551±0.007 | 0.448±0.028 | 0.887±0.004 |
| EM | 0.407±0.036 | 0.281±0.096 | 0.575±0.042 | 0.382±0.005 | 0.359±0.010 | 0.753±0.004 |
| EM-APL | 0.438±0.022 | 0.360±0.057 | 0.556±0.045 | 0.335±0.005 | 0.369±0.005 | 0.765±0.006 |
| SMILE | 0.242±0.014 | 0.146±0.037 | 0.462±0.003 | 0.308±0.007 | 0.328±0.004 | 0.628±0.003 |
| LL-R | 0.311±0.059 | 0.141±0.017 | 0.512±0.002 | 0.274±0.002 | 0.335±0.006 | 0.622±0.001 |
| LL-CP | 0.296±0.031 | 0.136±0.016 | 0.518±0.001 | **0.272±0.008** | 0.337±0.005 | 0.708±0.004 |
| LL-CT | 0.297±0.017 | 0.161±0.031 | 0.509±0.001 | 0.277±0.005 | 0.335±0.003 | 0.708±0.002 |
| CRISP | **0.164±0.012** | **0.082±0.018** | **0.455±0.002** | 0.276±0.002 | **0.324±0.001** | **0.620±0.001** |

Table 10: Predictive performance of each comparing methods on MLL datasets in terms of *Hamming loss* (mean ± std). The best performance is highlighted in bold (the smaller the better).

| | Image | Scene | Yeast | Corel5k | Mirflickr | Delicious |
|---|---|---|---|---|---|---|
| AN | 0.229±0.000 | 0.176±0.001 | 0.306±0.000 | **0.010±0.000** | 0.127±0.000 | **0.019±0.000** |
| AN-LS | 0.229±0.000 | 0.168±0.004 | 0.306±0.000 | **0.010±0.000** | 0.127±0.000 | **0.019±0.000** |
| WAN | 0.411±0.060 | 0.299±0.035 | 0.285±0.016 | 0.156±0.001 | 0.191±0.006 | 0.102±0.000 |
| EPR | 0.370±0.043 | 0.220±0.026 | 0.234±0.007 | 0.016±0.000 | 0.136±0.002 | 0.020±0.000 |
| ROLE | 0.256±0.018 | 0.176±0.017 | 0.279±0.010 | **0.010±0.000** | 0.128±0.000 | **0.019±0.000** |
| EM | 0.770±0.001 | 0.820±0.003 | 0.669±0.025 | 0.589±0.003 | 0.718±0.010 | 0.630±0.005 |
| EM-APL | 0.707±0.088 | 0.780±0.082 | 0.641±0.032 | 0.648±0.006 | 0.754±0.017 | 0.622±0.006 |
| SMILE | 0.219±0.009 | 0.182±0.021 | **0.208±0.002** | 0.010±0.000 | 0.127±0.001 | 0.081±0.008 |
| LL-R | 0.220±0.013 | 0.162±0.005 | 0.312±0.001 | 0.015±0.001 | 0.124±0.002 | **0.019±0.000** |
| LL-CP | 0.218±0.016 | 0.164±0.002 | 0.306±0.000 | 0.016±0.001 | 0.126±0.001 | **0.019±0.000** |
| LL-CT | 0.246±0.031 | 0.176±0.019 | 0.321±0.001 | 0.018±0.001 | 0.124±0.001 | **0.019±0.000** |
| CRISP | **0.165±0.023** | **0.140±0.013** | 0.211±0.001 | **0.010±0.000** | **0.121±0.002** | **0.019±0.000** |

## A.7 Details of Datasets

The details of the four MLIC datasets and the five MLL datasets are provided in Table 6 and Table 7 respectively. The basic statics about the MLIC datasets include the number of training set, validation set, and testing set (#Training, #Validation, #Testing), and the number of classes (#Classes). The basic statics about the MLL datasets include the number of examples (#Examples), the dimension of features (#Features), the number of classes (#Classes), and the domain of the dataset (#Domain).

## A.8 More Results of MLL Datasets

Table 8, 9, 10 and 11 report the results of our method and other comparing methods on five MLL datasets in terms of *Average Precision*, *Coverage, Hamming loss* and *One Error* respectively.

Table 11: Predictive performance of each comparing methods on MLL datasets in terms of *One-error* (mean ± std). The best performance is highlighted in bold (the smaller the better).

|  | Image | Scene | Yeast | Corel5k | Mirflickr | Delicious |
|---|---|---|---|---|---|---|
| AN | 0.708±0.096 | 0.626±0.123 | 0.489±0.194 | 0.758±0.002 | 0.358±0.005 | 0.410±0.012 |
| AN-LS | 0.643±0.052 | 0.578±0.111 | 0.495±0.130 | 0.736±0.009 | 0.360±0.015 | 0.454±0.013 |
| WAN | 0.670±0.060 | 0.543±0.060 | 0.239±0.002 | 0.727±0.012 | 0.352±0.010 | 0.404±0.002 |
| EPR | 0.703±0.046 | 0.615±0.090 | 0.240±0.003 | 0.764±0.000 | 0.362±0.015 | 0.441±0.008 |
| ROLE | 0.605±0.041 | 0.507±0.066 | 0.244±0.005 | 0.705±0.016 | 0.525±0.072 | 0.594±0.006 |
| EM | 0.769±0.036 | 0.681±0.119 | 0.326±0.079 | 0.656±0.009 | 0.365±0.008 | 0.446±0.009 |
| EM-APL | 0.773±0.045 | 0.812±0.059 | 0.341±0.109 | 0.690±0.007 | 0.434±0.023 | 0.405±0.006 |
| SMILE | 0.533±0.036 | 0.466±0.117 | 0.250±0.012 | 0.650±0.008 | 0.340±0.010 | **0.402±0.005** |
| LL-R | 0.597±0.084 | 0.490±0.054 | 0.436±0.087 | 0.715±0.006 | 0.342±0.016 | 0.543±0.041 |
| LL-CP | 0.629±0.043 | 0.450±0.051 | 0.240±0.000 | 0.731±0.016 | 0.357±0.016 | 0.490±0.028 |
| LL-CT | 0.616±0.019 | 0.574±0.074 | 0.552±0.097 | 0.726±0.022 | 0.375±0.012 | 0.475±0.019 |
| CRISP | **0.325±0.026** | **0.311±0.047** | **0.227±0.004** | **0.646±0.006** | **0.295±0.009** | **0.402±0.003** |

Table 12: Summary of the Wilcoxon signed-ranks test for CRISP against other comparing approaches at 0.05 significance level. The *p*-values are shown in the brackets.

| CRISP against | AN | AN-LS | WAN | EPR | ROLE | EM | EM-APL | SMILE | LL-R | LL-CP | LL-CT |
|---|---|---|---|---|---|---|---|---|---|---|---|
| *Coverage* | win[0.0313] | win[0.0313] | win[0.0313] | win[0.0313] | win[0.0313] | win[0.0313] | win[0.0313] | win[0.0313] | win[0.0431] | win[0.0431] | win[0.0431] |
| *One-error* | win[0.0313] | win[0.0313] | win[0.0313] | win[0.0313] | win[0.0313] | win[0.0313] | win[0.0313] | win[0.0431] | tie[0.625] | win[0.0313] | win[0.0313] |
| *Ranking loss* | win[0.0313] | win[0.0313] | win[0.0313] | win[0.0313] | win[0.0313] | win[0.0313] | win[0.0313] | win[0.0313] | win[0.0313] | win[0.0313] | win[0.0313] |
| *Hamming loss* | tie[0.0679] | tie[0.0679] | win[0.0313] | win[0.0313] | tie[0.0679] | win[0.0313] | win[0.0313] | tie[0.0796] | win[0.0313] | win[0.0313] | win[0.0313] |
| *Average precision* | win[0.0313] | win[0.0313] | win[0.0313] | win[0.0431] | win[0.0313] | win[0.0313] | win[0.0313] | tie[0.0938] | win[0.0313] | win[0.0313] | win[0.0313] |

Table 13: Predictive performance of CRISP compared with the approach of estimating priors from the validation set (CRISP-VAL) on the MLL datasets for five metrics.

|  | Metrics | Image | Scene | Yeast | Corel5k | Mirflickr | Delicious |
|---|---|---|---|---|---|---|---|
| CRISP | *Coverage* | **0.164±0.012** | **0.082±0.018** | **0.455±0.002** | **0.276±0.002** | **0.324±0.001** | **0.620±0.001** |
|  | *Ranking Loss* | **0.164±0.027** | **0.112±0.021** | **0.164±0.001** | **0.113±0.001** | **0.118±0.001** | **0.122±0.000** |
|  | *Average Precision* | **0.749±0.037** | **0.795±0.031** | **0.758±0.002** | **0.304±0.003** | **0.628±0.003** | **0.319±0.001** |
|  | *Hamming Loss* | **0.165±0.023** | **0.140±0.013** | **0.211±0.001** | **0.010±0.000** | **0.121±0.002** | **0.019±0.000** |
|  | *OneError* | **0.325±0.026** | **0.311±0.047** | **0.227±0.004** | **0.646±0.006** | **0.295±0.009** | **0.402±0.003** |
| CRISP-VAL | *Coverage* | 0.193±0.009 | 0.109±0.012 | 0.456±0.004 | 0.280±0.002 | 0.330±0.001 | 0.623±0.002 |
|  | *Ranking Loss* | 0.198±0.016 | 0.116±0.013 | 0.165±0.001 | 0.114±0.002 | 0.120±0.001 | **0.122±0.000** |
|  | *Average Precision* | 0.725±0.004 | 0.790±0.028 | 0.753±0.006 | 0.294±0.008 | 0.622±0.001 | **0.319±0.001** |
|  | *Hamming Loss* | 0.180±0.006 | 0.141±0.014 | 0.216±0.000 | **0.010±0.000** | 0.124±0.001 | **0.019±0.000** |
|  | *OneError* | 0.395±0.071 | 0.359±0.050 | 0.246±0.021 | 0.666±0.008 | 0.314±0.003 | 0.444±0.001 |

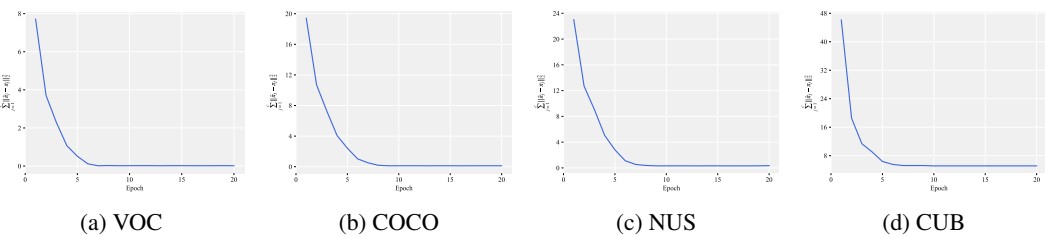

| (a) VOC | (b) COCO | (c) NUS | (d) CUB |
|---|---|---|---|

Figure 4: Convergence of $\hat{\pi}$ on four MLIC datasets.

## A.9  More Results of MLIC Datasets

Figure 4 illustrates the discrepancy between the estimated class-prior $\hat{\pi}_j$ and the true class-prior $\pi_j$ in every epoch on four MLIC datasets. During the initial few epochs, a significant decrease in the discrepancy between the estimated class-prior and the true class-prior is observed. After several epochs, the estimated class prior tends to stabilize and converges to the true class-prior. This result provides evidence that our proposed method effectively estimates the class-prior with the only observed single positive label.

## A.10  *p*-values of the wilcoxon signed-ranks test

Table 12 reports the *p*-values of the wilcoxon signed-ranks test [6] for the corresponding tests and the statistical test results at 0.05 significance level.

## A.11 Ablation results of MLL datasets

Table 13 reports the predictive performance of CRISP compared with the approach of estimating priors from the validation set (CRISP-VAL) on the MLL datasets for five metrics. The results show that CRISP outperforms CRISP-VAL on almost all the five metrics.

