# OpenReview forum: "Can Class-Priors Help Single-Positive Multi-Label Learning?"
_NeurIPS.cc/2025/Conference — NeurIPS 2025 poster_

### Official Review · Reviewer_wnkV · 2025-06-02

**Clarity:** 2
**Significance:** 2
**Originality:** 3
**Rating:** 4
**Confidence:** 4

**Summary:**

The authors propose a novel framework that addresses a key limitation in Single-Positive Multi-Label Learning: the unrealistic assumption that all classes have identical prior probabilities. The method estimates class priors using a threshold-based approach with theoretical convergence guarantees, then derives an unbiased risk estimator for classification. Extensive experiments demonstrate superior performance over existing SPMLL approaches.

**Questions:**

I have the following two questions. First, how sensitive is the class-prior estimation to the quality of the initial model predictions? Poor initial models might lead to unreliable prior estimates, creating a chicken-and-egg problem. Second, have the authors considered other class-prior estimation techniques beyond the threshold-based approach? Comparison with alternative strategies would strengthen the contribution.

**Ethical Concerns:**

["NO or VERY MINOR ethics concerns only"]

**Final Justification:**

I have reviewed the authors' rebuttal and discussions with other reviewers. The main concerns have been adequately addressed, and the authors provided convincing responses to the technical questions raised. I maintain my borderline accept recommendation.

**Limitations:**

the authors adequately addressed the limitations.

**Paper Formatting Concerns:**

No major formatting concerns identified.

**Quality:**

3

**Strengths And Weaknesses:**

Strengths：
The paper provides convergence guarantees for both class-prior estimation (Theorem 4.1) and risk minimization (Theorem 4.2). The theoretical analysis appears sound, though the assumptions are somewhat restrictive. And the consistent improvements across multiple datasets and metrics provide reasonable evidence of the method's effectiveness.

Weaknesses：
The warm-up strategy using AN solution seems necessary but lacks theoretical grounding. The related work section reads more like a literature survey than establishing clear connections to the proposed approach.

---

> ### Author Rebuttal · Authors · 2025-07-31
>
> Thank you for taking the time to review the paper and providing valuable feedback. I appreciate your efforts in ensuring the quality of the research. Regarding your concerns, I would like to provide the following explanations:
>
>
> > 1. The warm-up strategy using AN solution seems necessary but lacks theoretical grounding. How sensitive is the class-prior estimation to the quality of the initial model predictions? Poor initial models might lead to unreliable prior estimates, creating a chicken-and-egg problem.
>
> The warm-up strategy using AN loss is actually a well-established practice in the SPMLL, adopted by several classical algorithms [1-4]. To address your concern about sensitivity to initial model quality, we conducted an ablation study on VOC and COCO datasets, training the warm-up model for different numbers of epochs (1, 2, and 3) to simulate varying initial classifier quality:
>
> | **Epochs** | **AN Loss (VOC)** | **CRISP Final (VOC)** | **AN Loss (COCO)** | **CRISP Final (COCO)** |
> | :--------: | :---------------: | :-------------------: | :----------------: | :--------------------: |
> |     1      |       43.54       |         88.50         |       45.21        |         73.85          |
> |     2      |       76.25       |         89.10         |       56.33        |         74.30          |
> |     3      |       83.21       |         89.93         |       62.11        |         74.91          |
>
> The results demonstrate that even with a weak initial classifier (1 epoch, mAP of only 43.54% on VOC), CRISP achieves strong final performance (88.50%). The final performance remains stable across different warm-up qualities, with only minor improvements as the warm-up classifier gets stronger. This suggests that the class-prior estimation mechanism is sufficiently robust to work with imperfect initial predictions and progressively refine them during training.
>
> [1] Label-aware global consistency formulti-label learning with single positive labels
>
> [2] One positive label is sufficient: Single-positive multi-label learning with label enhancement.
>
> [3] Large loss matters in weakly supervised multi-label classification
>
> [4] revisiting pseudo label for single positive multi label learning
>
> > 2. The related work section reads more like a literature survey than establishing clear connections to the proposed approach.
>
> Thank you for this valuable feedback. We will revise the related work section in the next version to more clearly establish the connections between our proposed approach and existing methods.
>
> > 3. Have the authors considered other class-prior estimation techniques beyond the threshold-based approach? Comparison with alternative strategies would strengthen the contribution.
>
> We have actually compared our threshold-based class-prior estimation with an alternative strategy in our ablation study (Table 3). Specifically, we used the frequency of each labeled positive sample appearing in the validation set as the class prior. Our experimental results show that our approach outperforms the alternative strategy, demonstrating the effectiveness of our proposed method.
>
> We hope that our revisions and additional clarifications have fully addressed your concerns. Please feel free to let us know if further explanations or additional analyses are needed.

---

### Official Review · Reviewer_8tLt · 2025-06-08

**Clarity:** 2
**Significance:** 2
**Originality:** 3
**Rating:** 4
**Confidence:** 4

**Summary:**

This paper proposed method named CRISP for single positive multi-label learning. A class-priors estimator is introduced with   a convergence guarantee.  The author claims that the experimental results demonstrate the effectiveness and superiority of this method over existing approaches.

**Questions:**

- Please add more results comparison with the recent papers

- Why choose ranking loss? It would be better to add the commonly used metrics including mAP, F1, Recall, etc.

**Ethical Concerns:**

["NO or VERY MINOR ethics concerns only"]

**Final Justification:**

I have read the author's response and it has resolved all my mentioned concerns. So I will keep the score as borderline accepted.

**Limitations:**

Yes

**Quality:**

3

**Strengths And Weaknesses:**

Strength

- The single positive Multi-label learning is a pratical setting in the classification task since the labeling is usually expensive, it is worthy to spend effort on this task.

- It is great that the author provides theoretical guarantees on the convergence of the estimated class priors (to ground-truth class priors), which makes the approach convincing.


Weakness

-  the methods used for comparison are from several years ago, it would be better to make comparison with the latest papers on this topic.

For example,

[1]. Chen, Yanxi, et al. "Boosting single positive multi-label classification with generalized robust loss." Proceedings of the Thirty-Third International Joint Conference on Artificial Intelligence. 2024.

[2] Xing, Xin, et al. "Vision-Language Pseudo-Labels for Single-Positive Multi-Label Learning." Proceedings of the IEEE/CVF Conference on Computer Vision and Pattern Recognition. 2024.

[3]. Liu, Biao, et al. "Revisiting pseudo-label for single-positive multi-label learning." International Conference on Machine Learning. PMLR, 2023.

[4]. Ding, Zixuan, et al. "Exploring structured semantic prior for multi label recognition with incomplete labels." Proceedings of the IEEE/CVF Conference on Computer Vision and Pattern Recognition. 2023.

[5]. Kim, Youngwook, et al. "Large loss matters in weakly supervised multi-label classification." Proceedings of the IEEE/CVF Conference on Computer Vision and Pattern Recognition. 2022.

- One thing confused me is the metrics utilized for demonstrating your experiment results and make comparisons. Why choose ranking loss? It would be better to add the commonly used metrics including mAP, F1, Recall, etc.

---

> ### Author Rebuttal · Authors · 2025-07-31
>
> Thank you for taking the time to review the paper and providing valuable feedback. I appreciate your efforts in ensuring the quality of the research. Regarding your concerns, I would like to provide the following explanations:
>
> For Weaknesses:
>
> > 1. the methods used for comparison are from several years ago, it would be better to make comparison with the latest papers on this topic.
>
> Thank you for the valuable suggestions. We note that [5] (LL) has already been included as one of our baselines. In the revised version, we additionally include [1] and [3] as new baselines, and the comparison results are shown below. Our method achieves superior performance on most datasets, with only a slight gap compared to MIME on the CUB dataset. Regarding [2] and [4], both methods rely on CLIP-based pseudo-labeling, which introduces extra supervision information beyond single-positive labels. Direct comparison with our method may therefore be unfair. However, we will discuss these approaches in the Related Work section to provide a more comprehensive overview.
>
> | **Method** |       VOC        |       COCO       |       NUS        |       CUB        |
> | :--------: | :--------------: | :--------------: | :--------------: | :--------------: |
> |    MIME    |   89.20 ± 0.16   |   72.92 ± 0.26   |   48.74 ± 0.43   | **21.89 ± 0.35** |
> |  GR Loss   |   89.42 ± 0.23   |   73.15 ± 0.22   |   49.09 ± 0.02   |   21.36 ± 0.12   |
> |   CRISP    | **89.82 ± 0.19** | **74.64 ± 0.22** | **50.00 ± 0.32** |   21.65 ± 0.18   |
>
> > 2. One thing confused me is the metrics utilized for demonstrating your experiment results and make comparisons. Why choose ranking loss? It would be better to add the commonly used metrics including mAP, F1, Recall, etc.
>
> Thank you for the suggestion. In fact, we have already evaluated our method using multiple commonly used metrics, including mAP, Ranking Loss, Hamming Loss, One-Error, and Coverage, as shown in Tables 1, 2, 8, 9, 10, and 11. These metrics jointly assess different aspects of model performance, such as ranking quality, per-label accuracy, and top prediction correctness. We will clarify this in the revised version to avoid confusion.
>
> We hope that our revisions and additional clarifications have fully addressed your concerns. Please feel free to let us know if further explanations or additional analyses are needed.

---

> > ### Comment · Reviewer_8tLt · 2025-08-01
> >
> > Thanks for the clarification! My concerns are well addressed.

---

### Official Review · Reviewer_5pXo · 2025-06-25

**Clarity:** 3
**Significance:** 2
**Originality:** 2
**Rating:** 4
**Confidence:** 4

**Summary:**

This paper presents a statistical strategy for building stronger supervision signals in weakly-annotated settings by estimating class probabilities with the model which is in the process of being trained, and samples from the data.

**Questions:**

Here I include both direct questions, and comments to which I hope you will respond as appropriate.
1. l. 116: "ratio between the fraction
of the total number of samples and that of positive labeled samples receiving scores above the threshold, thereby obtaining the class-prior probability of the j-th label": this was pretty unclear.
2. l. 123: empirical estimator $\hat{q}$: since this is estimating an expectation by an un-weighted sum, you need to describe the sampling of $x_i$ from $\mathcal X$ for this estimator.
3. eqs 6, 7: the class priors you are using for supervision can not be known in ground-truth. Are you in fact using the estimations as per Algorithm 1?
4.  you never demonstrate that $\pi_i = p(y_i=1)$
5. eq 4: you should mention that this is binary cross-entropy. In general, your thresholded classifier approach only makes sense in contexts where the classifier's outputs can be interpreted as a confidence score for each class. A classifier trained on a hinge loss, for example, would not fit this framework.
6. the "class-prior" vocabulary implies a Bayesian framework, but the notion of updating priors based on data is contradictory to the Bayesian viewpoint, where the prior is fixed a priori. Bayesian data-based updates are typically made on the posterior.
7. why do you split data differently for MLL and MLIC datasets?
8. Why are separate statistics being reported for MLL and MLIC (tables 6 and 7). I'd like to know table 6 and 7's reported statistics on both the MLIC and MLL datasets.
9. Naively, we could estimate $p(y_i=1)$ by looking at the data only and counting the number of times the label $y_i$ is positive and dividing by the total number of observations. Why is the proposed method, which is more of a pseudo-signal since it is dependent on the classifier, better? Empirical evidence here would be most convincing.
10. l. 240: "Without the constraint of the class-priors, the predicted class-prior probability diverges from the true class-prior as epochs increase, significantly impacting the model’s performance." The claim that model performance is impacted by this phenomenon is not sufficiently supported. Predicted and true probabilities do not necessarily have to be close for the correct discrete decision to be made. The issue of accurate probability prediction is one of model calibration, which has its own specialized performance metrics for this explicit reason; standard performance metrics are often agnostic to calibration.
11. Figure 1: how do you compute the predicted class-priors for these methods. Presumably, they do not all include explicit class-prior prediction strategies in their published methodologies.

**Ethical Concerns:**

["NO or VERY MINOR ethics concerns only"]

**Final Justification:**

My primary concerns during this discussion process were the use of the word "prior", the similar performance between the proposed method and naive sampling, and some experimental procedure related to the estimation of other methods' marginal class probability estimates.

The authors agreed to shift the text away from "class-priors".

The authors pointed out that the naive-sampling ablation was done over the fully labelled validation set and is not a practical solution to the problem, since the available data during model training is mostly un-labelled. They pointed out that estimating class marginal probabilities by taking the frequency of labels on the train set will be biased, demonstrating theoretically. I asked them to show empirical results to demonstrate this.

While the authors did not improve their estimation of other methods' marginal class probability estimates, their clarified procedure is, while not generous, adequate considering it is not a primary result.

**Limitations:**

The authors make brief discussion of time complexity. The discussion of time-complexity as a limitation is possible in nearly any modern machine-learning work, and in my opinion does not constitute an earnest attempt to address a work's limitations.

**Paper Formatting Concerns:**

No issues.

**Quality:**

2

**Strengths And Weaknesses:**

Strengths
1. The paper is laid out in a mostly clear way, with an obvious motivation/story
2. Sensitivity studies are presented and statistical significances are computed
3. while the notion of supervising multi-label classification training with dataset-level class-frequency statistics is not in itself novel, the presented approach to this end is novel (although as yet not shown to be especially impactful relative to even naive options).

Weaknesses
1. A crucial step is missing from the method, connecting the ratio $\pi$ to the class probability.
2. Some experimental details are missing, as referenced in the comments/questions below

---

> ### Author Rebuttal · Authors · 2025-07-31
>
> Thank you for taking the time to review the paper and providing valuable feedback. I appreciate your efforts in ensuring the quality of the research. Regarding your concerns, I would like to provide the following explanations:
>
> > 1. l. 116: "ratio between the fraction of the total number of samples and that of positive labeled samples receiving scores above the threshold, thereby obtaining the class-prior probability of the j-th label": this was pretty unclear.
>
> What we intended to express is: For label j, we calculate the class-prior $\pi_j$ as the ratio between the total number of samples and the number of samples with scores above threshold $\hat z$. For example, if we have 1000 samples in total and 300 samples have scores above $\hat z$, then the estimated class-prior $\pi_j = 300/1000 = 0.3$​. We will clarify this in the revision.
>
> > 2. l. 123: empirical estimator : since this is estimating an expectation by an un-weighted sum, you need to describe the sampling of from for this estimator.
>
> The samples $\boldsymbol{x}_i$ in our empirical estimator are independently and identically distributed (i.i.d.) samples drawn from the marginal distribution $p(\boldsymbol{x})$. We will add this clarification to make the estimator more rigorous.
>
> > 3. eqs 6, 7: the class priors you are using for supervision can not be known in ground-truth. Are you in fact using the estimations as per Algorithm 1?
>
> We confirm that the class priors used in Eqs. (6) and (7) are indeed the estimates obtained via Algorithm 1.
>
> > 4. you never demonstrate that $\pi_i = p(y_i=1)$
>
> While we have described $\pi_i$ as the class prior probability in the text, we acknowledge that we should have explicitly stated the mathematical relationship $\pi_i = p(y_i=1)$. We will add this explicit equation in the revised version.
>
> > 5. eq 4: you should mention that this is binary cross-entropy. In general, your thresholded classifier approach only makes sense in contexts where the classifier's outputs can be interpreted as a confidence score for each class. A classifier trained on a hinge loss, for example, would not fit this framework.
>
> We would like to clarify that our method uses L1 loss (absolute loss) as stated in Eq. (5). We agree that the loss should allow the classifier output to be interpreted as a confidence score, which can be achieved by applying a sigmoid function to the logits.
>
> > 6. the "class-prior" vocabulary implies a Bayesian framework, but the notion of updating priors based on data is contradictory to the Bayesian viewpoint, where the prior is fixed a priori. Bayesian data-based updates are typically made on the posterior.
>
> We clarify that in our setting the true class-prior probabilities are indeed fixed, consistent with the Bayesian framework. What we update during training is the estimate of the class-priors, not the true priors themselves. As the classifier improves, the estimated class-priors become more accurate, and this alternated optimization leads to better performance.
>
> > 7. why do you split data differently for MLL and MLIC datasets?
>
> For the MLIC datasets, the train/test splits are predefined, so we simply hold out a portion of the training set for validation. In contrast, the MLL datasets do not have predefined splits, so we manually split them into train/validation/test sets. This difference in splitting strategy does not affect the fairness of comparison.
>
> > 8. Why are separate statistics being reported for MLL and MLIC (tables 6 and 7). I'd like to know table 6 and 7's reported statistics on both the MLIC and MLL datasets.
>
> As mentioned above, the MLIC datasets are pre-defined **image** datasets, so we report their statistics in terms of #Training / #Validation / #Testing / #Classes. In contrast, the MLL datasets are **tabular** datasets without predefined splits, so we report their statistics as #Examples / #Features / #Classes.
>
> > 9. Naively, we could estimate by looking at the data only and counting the number of times the label is positive and dividing by the total number of observations. Why is the proposed method, which is more of a pseudo-signal since it is dependent on the classifier, better? Empirical evidence here would be most convincing.
>
> In SPMLL, each instance is associated with only one observed positive label, while the remaining labels are unobserved. As a result, a large portion of positive labels is missing, making it impossible to accurately estimate class priors by simply counting the observed positive labels and dividing by the total number of samples. To address this challenge, our method iteratively refines class-prior estimates by leveraging both the classifier outputs and the limited available labels.. Moreover, we provide empirical evidence in the ablation study (Table 3), where we used the frequency of labeled positive samples in the validation set as the class-prior (“CRISP-VAL”). The results show that our proposed method achieves better performance than this naive approach, because our estimator leverages both the classifier’s outputs and the available labels to iteratively refine the class-prior estimates.
>
> > 10. l. 240: "Without the constraint of the class-priors, the predicted class-prior probability diverges from the true class-prior as epochs increase, significantly impacting the model’s performance." The claim that model performance is impacted by this phenomenon is not sufficiently supported. Predicted and true probabilities do not necessarily have to be close for the correct discrete decision to be made. The issue of accurate probability prediction is one of model calibration, which has its own specialized performance metrics for this explicit reason; standard performance metrics are often agnostic to calibration.
>
> We will revise this statement in the next version. Our intention was to suggest that lack of class-prior alignment may be one factor contributing to poorer performance, rather than claiming a direct causal relationship. Nonetheless, our experiments clearly show that our method achieves better alignment between predicted and true class-priors.
>
> > 11. Figure 1: how do you compute the predicted class-priors for these methods. Presumably, they do not all include explicit class-prior prediction strategies in their published methodologies.
>
> For all methods, we compute the predicted class-priors as the average predicted confidence score for each class across all samples. Even though most baselines do not explicitly estimate class-priors, their model outputs can still be interpreted as confidence scores, so taking the mean over samples provides a consistent way to compare predicted class-priors across methods.
>
> We hope that our revisions and additional clarifications have fully addressed your concerns. Please feel free to let us know if further explanations or additional analyses are needed.

---

> > ### Comment · Reviewer_5pXo · 2025-08-03
> >
> > 1. Fine, I was more so pointing out that the text should be edited in the paper to be clearer, not asking for personal clarification. Typically, when we describe a ratio, ``the fraction between [numerator] and [denominator].
> > 2. OK. I think adding that in practice you get samples from p(x) directly in the form of a dataset, which you assume is i.i.d sampled from an underlying distribution which you do not know.
> > 3. OK. I think you should spell that out in the text. It should only take a line.
> > 4. My mistake, this was indeed mentioned in the text.
> > 5. In the preamble to eq. 5 at line 148 you should mention the loss you use explicitly. I see that it is mentioned in the appendix link in the footnote, but this will cost you no space, as the line at 149 ends early. In response to "which can be achieved by applying a sigmoid function to the logits": while the application of a sigmoid can achieve a [0,1] output which can mechanically be applied as a confidence, this interpretation is only reasonable given a certain category of losses and model architectures. Going back to the hinge loss, this loss, and its associated models, result in model outputs which are almost always (extremely close to) -1 or 1. There is no built-in notion of relative score, confidence, or class probability.
> > 6. I remain unconvinced. I think maybe the authors are confusing the likelihood p(c | x) marginalized over x to be p(c) with the prior because they are distributions over the same random variable. There is no notion of "estimating the true prior" in Bayesian statistics. The prior is for the practitioner to inject some a-priori problem-specific knowledge.
> > 7. How did you choose the split? The most obvious default-choice would be to replicate the ratio-wise split of the other dataset.
> > 8. Yes, but you imposed splits, meaning that you can report your #Examples split into train/validation/test for the tabular dataset. Similarly, it wouldn't hurt to just put 3 under number of features for the image dataset (assuming it's a 3-channel image).
> > 9. The performance reported in the CRISP-val ablation shows extremely marginal performance improvement over a method which you describe in your response as impossible ("a large portion of positive labels is missing, making it impossible to accurately estimate class priors by simply counting the observed positive labels and dividing by the total number of samples"). Would the naive estimate likely not become even better if the much larger training set were used to estimate the priors? I think it makes more sense to use the training set since you are setting parameters for your estimation.
> > 10. OK.
> > 11. Fine. I think it would make more sense to do this over the indicator of predictions, considering that the competitor methods may not be explicitly designed for calibration.

---

> > > ### Author Response · Authors · 2025-08-04
> > >
> > > We thank the reviewer for the continued feedback and appreciate the opportunity to clarify our terminology and intent:
> > >
> > > 1. Thank you for the suggestion. We will revise the wording in the paper to make the description clearer in the final version.
> > >
> > > 2. We will explicitly add this assumption in the revision, stating that the samples $\\{\boldsymbol{x} _i\\} _{i=1}^n$ is assumed to consist of i.i.d. samples drawn from an unknown underlying marginal distribution $p(\boldsymbol x)$.
> > >
> > > 3. We will explicitly state in the revision that the class-priors used in Eqs. (6) and (7) are the estimates obtained via Algorithm 1.
> > >
> > > 4. Thank you for pointing this out, and we appreciate your careful reading of our paper.
> > >
> > > 5. Thank you for the detailed suggestion. We will explicitly state the loss function we use in the preamble to Eq. (5) in the revised version. Regarding the interpretation of outputs as confidence scores, we will clarify this limitation in the revision. Our method is indeed not applicable to loss functions such as the hinge loss. However, with the current model architecture and the use of L1 loss, our experiments demonstrate that the proposed method is reasonable and effective.
> > >
> > > 6. Thank you for your continued clarification. We agree that our use of the term *class-prior* may cause confusion from a Bayesian perspective. The $\pi_j = p(y_j = 1)$ in our work refers to the marginal class probability under the data distribution. In the revised version, we will clarify this distinction and consider replacing “class-prior” with more precise terminology such as "class marginal probability" to avoid confusion.
> > >
> > > 7. As stated in lines 192–194, for the MLIC datasets, we follow the predefined train/test splits provided by the original datasets. We use the original test set for evaluation and randomly hold out 20% of the training set as the validation set. For the MLL datasets, since no official splits are provided, we randomly split the data into 80% for training, 10% for validation, and 10% for testing.
> > >
> > > 8. Thank you for the suggestion. In the revised version, we will integrate all relevant dataset statistics into a unified table, as shown below. For the MLIC datasets, we add the #Features (set to 3 for RGB images) and #Domain (“Images”). For the MLL datasets, we include #Training / #Validation / #Testing, based on an 80%/10%/10% split of the total examples.
> > >
> > >    | Dataset   | #Training | #Validation | #Testing | #Features             | #Classes | Domain  |
> > >    | --------- | --------- | ----------- | -------- | --------------------- | -------- | ------- |
> > >    | VOC       | 4574      | 1143        | 5823     | $3\times448\times448$ | 20       | Images  |
> > >    | COCO      | 65665     | 16416       | 40137    | $3\times448\times448$ | 80       | Images  |
> > >    | NUS       | 120000    | 30000       | 60260    | $3\times448\times448$ | 81       | Images  |
> > >    | CUB       | 4795      | 1199        | 5794     | $3\times448\times448$ | 312      | Images  |
> > >    | Image     | 1600      | 200         | 200      | 294                   | 5        | Images  |
> > >    | Scene     | 1925      | 241         | 241      | 294                   | 6        | Images  |
> > >    | Yeast     | 1933      | 242         | 242      | 103                   | 14       | Biology |
> > >    | Corel5k   | 4000      | 500         | 500      | 499                   | 374      | Images  |
> > >    | Mirflickr | 19665     | 2458        | 2458     | 1000                  | 38       | Images  |
> > >    | Delicious | 12871     | 1610        | 1610     | 500                   | 983      | Text    |
> > >
> > > 9. In the SPMLL setting, each training instance is associated with only one observed positive label, while all other labels are unobserved. This makes it impossible to accurately estimate class-prior frequencies from the training set, since most true positives remain hidden. If we were to use the frequency of observed labels in the training set directly, it would implicitly introduce additional supervision, as it assumes access to aggregated label statistics beyond what SPMLL provides. Instead, we use the validation set—where full labels are available—for a controlled ablation (CRISP-VAL).
> > >
> > > 10. Thanks for your suggestion.
> > >
> > > 11. We chose to use the raw predicted confidence scores to avoid introducing additional hyperparameters related to threshold selection, which can vary across methods and affect fairness.
> > >
> > > We hope that our revisions and additional clarifications have fully addressed your concerns. Please feel free to let us know if further explanations or additional analyses are needed.

---

> > > > ### Comment · Reviewer_5pXo · 2025-08-05
> > > >
> > > > 9. If we assume that labels are masked uniformly at random in the train-set to build the SPMLL setting, measuring the frequency over just observed labels would be a good approximation of actual label frequency, assuming enough samples. Is the uniform assumption valid? If not, why?
> > > >
> > > > If there are enough samples, then your claim that naive measurement is impossible. If there are not enough samples, then your CRISP-VAL results when computed over the observed train-set would demonstrate the truth of this claim and go far to motivate your work. As the current CRISP-VAL results stand currently, they discredit your work because the naive frequency measurement version CRISP-VAL is very close in performance to CRISP.
> > > >
> > > > All other points have been addressed.

---

> > > > > ### Author Response · Authors · 2025-08-07
> > > > >
> > > > > Dear Reviewer,
> > > > >
> > > > > We greatly appreciate the time and effort you've taken to review our submission. We hope that our response has addressed your concerns, and we look forward to your feedback. Please let us know if you need any further information or clarification. We are fully open to engaging in further discussions to improve our work.
> > > > >
> > > > > Best regards,
> > > > >
> > > > > Authors

---

> ### Author Response · Authors · 2025-08-06
>
> Thank you for your continued feedback. We would like to provide the following clarifications:
>
> > If we assume that labels are masked uniformly at random in the train-set to build the SPMLL setting ...
>
> Our method does not require the assumption that single-positive labels are sampled uniformly at random. Even under non-uniform selection of the single positive label, our class marginal probabilities estimation framework remains valid. On the other hand, even under the uniform sampling assumption, estimating the true class marginal probabilities by directly counting the frequency of observed labels in train-set is still biased.
>
> This is because the probability $p(l ^ k = 1)$, i.e., label $k$ being selected as the single positive label, is not equal to $p(y ^ k = 1)$, but rather proportionally biased downward, due to the uniform sampling over the set of positive labels. We include the derivation below to illustrate this point:
>
> Let $\boldsymbol{y} \in \{0,1\} ^ c$ be the binary label vector with c classes, and let $l ^ k = 1$ denote that class $k$ is selected as the observed single-positive label. If the positive label is sampled uniformly from the set $\{ j : y ^ j = 1 \}$, then for any $k$ such that $y ^ k = 1$, the probability that it is selected is:
>
> $$
> p(l^k = 1 \mid \boldsymbol{y},y^k=1) = \frac{1}{\sum_{j=1}^c \mathbb{I}(y^j = 1)}
> $$
>
>
> Then, marginalizing over $\boldsymbol{y}$, we get:
>
> $$
> p(l^k = 1) = \int p(l^k = 1 \mid \boldsymbol{y})  p(\boldsymbol{y})  d\boldsymbol{y}
> $$
>
> The term $p(l ^ k = 1 \mid \boldsymbol{y})$ can be decomposed as follows:
>
> $$
> p(l ^ k = 1 \mid \boldsymbol{y}) = p(l ^k = 1 \mid \boldsymbol{y},y ^k=0)p(y ^k=0|\boldsymbol{y}) \\ + p(l ^k = 1 \mid \boldsymbol{y},y ^k=1)p(y ^k=1|\boldsymbol{y})
> $$
>
> Given that $y ^ k = 0$ implies the $k$-th label cannot be selected as the single positive label, the term $p(l ^ k = 1 \mid \boldsymbol{y}, y ^ k = 0)$ becomes zero. Thus, the expression simplifies to:
> $$
> p(l^k = 1 \mid \boldsymbol{y}) = p(l^k = 1 \mid \boldsymbol{y},y^k=1)p(y^k=1|\boldsymbol{y})
> $$
> Substituting this into $p(l ^ k = 1) = \int p(l ^ k = 1 \mid \boldsymbol{y})  p(\boldsymbol{y}) d\boldsymbol{y}$：
> $$
> p(l ^ k = 1) = \int p(l ^ k = 1 \mid \boldsymbol{y})  p(\boldsymbol{y})  d\boldsymbol{y} \\
> $$
> $$
> = \int p(l ^k = 1 \mid \boldsymbol{y},y ^k=1)p(y ^k=1|\boldsymbol{y})  p(\boldsymbol{y})  d\boldsymbol{y} \\
> $$
> $$
> = \int \frac{1}{\sum _{j=1} ^c \mathbb{I}(y ^j = 1)} \cdot p(\boldsymbol{y}|y ^k = 1) \cdot p(y ^k=1) d\boldsymbol{y} \\
> $$
> $$
> = p(y ^k = 1) \cdot \mathbb{E} _{\boldsymbol{y} \sim p(\boldsymbol{y}|y ^k=1)} \left[ \frac{1}{\sum _{j=1}^c \mathbb{I}(y ^j = 1)} \right]
> $$
> This gives:
> $$
> p(l^k = 1) = \alpha \cdot p(y^k = 1)
> $$
> where $\alpha = \mathbb{E} _ {\boldsymbol{y} \sim p(\boldsymbol{y}|y ^ k=1)} \left[ \frac{1}{\sum _ {j=1} ^ c \mathbb{I}(y ^ j = 1)} \right]$. Thus, the frequency of observing $l ^ k = 1$ under uniform sampling is a biased estimate of $p(y ^ k = 1)$, and must be re-scaled by $1/\alpha$ to recover an unbiased estimate. However, the scaling factor is itself nontrivial to estimate, which makes naive frequency-based methods on train-set unreliable in practice.
>
> > If there are enough samples, then your claim that naive measurement is impossible. If there are not enough samples, then your CRISP-VAL results when ...
>
> 1. Our core motivation lies in addressing the class imbalance challenge in SPMLL by proposing a general framework that combines a class marginal probability estimator with a tailored loss function. Both our method (CRISP) and the frequency-based estimator using the validation set (CRISP-VAL) achieve competitive performance, which validates the effectiveness of our framework regardless of the estimation strategy.
>
> 2. Regarding the concern that CRISP-VAL performs close to CRISP, we believe this is because we held out 20% of the training data as validation, which provides a relatively large and clean set to estimate the class marginal probabilities $p(y ^ j = 1)$. However, in real-world SPMLL scenarios, obtaining such a well-annotated validation set is costly and often unrealistic.
>
>    To further demonstrate this, we conducted an additional experiment using only 5% of the training data as the validation set to estimate the class-priors. The results show a notable performance drop, confirming that frequency-based estimates degrade rapidly when fewer observed labels are available, while our method is robust.
>
> | **Dataset** | **CRISP-VAL (5%)** |     **CRISP**      |
> | :---------: | :----------------: | :----------------: |
> |     VOC     |   86.314 ± 0.201   | **89.820 ± 0.191** |
> |    COCO     |   71.664 ± 0.236   | **74.640 ± 0.219** |
>
> These results further motivate the need for an accurate, data-efficient estimator like ours in low-annotation regimes.
>
> We hope that our revisions and additional clarifications have fully addressed your concerns. Please feel free to let us know if further explanations or additional analyses are needed.

---

> ### Comment · Reviewer_5pXo · 2025-08-07
>
> This mostly resolves my concern. I will raise my score to 4: Borderline Accept.
>
>
> I have one final clarification to ask, related to the previous response:
> When you say you take 5% of the training data as validation set for the new version of CRISP-VAL results you show, are you using the pre-SPMLL-masked labels or the post-masked data.
>
> I understand the point you make that sampling from the masked data biases frequency estimates, but it would be useful to see to what magnitude this bias occurs in practice.

---

> > ### Author Response · Authors · 2025-08-08
> >
> > Thank you very much for your recognition and for raising the score.
> >
> > To clarify your question: the 5% of the training data we used as a validation set in CRISP-VAL refers to the labels before applying the SPMLL masking — thus, labels are accurate and complete.
> >
> > Additionally, as you suggested, we further conducted experiments where we estimate the class marginal distribution using the frequency of masked training labels, The results are as follows:
> >
> > | Dataset | CRISP (masked label freq) |
> > | ------- | ------------------------- |
> > | VOC     | 83.112 ± 0.106            |
> > | COCO    | 70.671 ± 0.093            |
> >
> > We observe that this naive estimation leads to clear performance degradation.

---

### Official Review · Reviewer_GgxX · 2025-07-01

**Clarity:** 3
**Significance:** 2
**Originality:** 3
**Rating:** 4
**Confidence:** 4

**Summary:**

The focal issue addressed in this paper is single-positive multi-label learning (SPMLL), a variant of the multi-label learning paradigm under weak supervision where each example is labeled with only one positive label, leaving other labels unannotated. This scenario poses a challenge because conventional SPMLL methodologies make the unrealistic assumption that all class priors are identical, which doesn't hold true in practical applications where class-prior distributions can vary significantly. To address this limitation, the paper introduces a novel framework named CRISP (Class-pRiors Induced Single-Positive multi-label learning). CRISP encompasses a class-priors estimator designed to theoretically converge to the actual class-priors of the data. Furthermore, leveraging these estimated class-priors, CRISP formulates an unbiased risk estimator for classification tasks. This estimator ensures that the corresponding risk minimizer is approximately equivalent to the optimal risk minimizer achievable with fully supervised data.

**Questions:**

1.Will the inclusion of the additional absolute operator of Equation 7 compromise the unbias of the proposed risk estimator?
2.Why weren’t BOOSTLU and PLC compared on the MLL datasets?

**Ethical Concerns:**

["NO or VERY MINOR ethics concerns only"]

**Final Justification:**

Overall a solid work. My previous concerns on the affect of warm-up classifier and missing baselines have been partially addressed.

**Limitations:**

yes

**Quality:**

3

**Strengths And Weaknesses:**

Strengths
1.The paper theoretically guarantees that the estimated class-priors will converge to the true class-priors. Furthermore, based on these class-priors, the paper introduces a consistent risk estimator that ensures the corresponding risk minimizer will approximately converge to the optimal risk minimizer on fully supervised data. The solid theoretical foundation lends credibility to the proposed approach.
2.The paper presents an extensive experimental validation, demonstrating the necessity of addressing class-prior issues and the effectiveness of the proposed method through a comprehensive set of experiments, including ablation studies designed to validate the effectiveness of the proposed class-priors estimator and an analysis of the time complexity.
Weaknesses
1.The effectiveness of the method appears to rely in part on the performance of the warm-up classifier; if the initial classifier is too weak, it may undermine the reliability of thresholding and lead to degraded practical performance.
Some recent baselines are missing, such as “Exploring Structured Semantic Prior for Multi-Label Recognition With Incomplete Labels.

---

> ### Author Rebuttal · Authors · 2025-07-31
>
> Thank you for taking the time to review the paper and providing valuable feedback. I appreciate your efforts in ensuring the quality of the research. Regarding your concerns, I would like to provide the following explanations:
>
> For weaknesses:
>
> > 1. The effectiveness of the method appears to rely in part on the performance of the warm-up classifier; if the initial classifier is too weak, it may undermine the reliability of thresholding and lead to degraded practical performance.
>
> To evaluate how much our method depends on the quality of the warm-up classifier, we conducted an additional ablation study on VOC and COCO. Specifically, we trained the warm-up model using the AN loss for 1, 2, and 3 epochs to simulate classifiers with varying initial quality. The results are shown below:
>
> | **Epochs** | **AN Loss (VOC)** | **CRISP Final (VOC)** | **AN Loss (COCO)** | **CRISP Final (COCO)** |
> | :--------: | :---------------: | :-------------------: | :----------------: | :--------------------: |
> |     1      |       43.54       |         88.50         |       45.21        |         73.85          |
> |     2      |       76.25       |         89.10         |       56.33        |         74.30          |
> |     3      |       83.21       |         89.93         |       62.11        |         74.91          |
>
> As shown, our method remains stable even when the warm-up classifier is relatively weak. This suggests that while the initial classifier provides a starting point for thresholding, the overall method is robust to its performance within a reasonable range.
>
> > 2. Some recent baselines are missing, such as “Exploring Structured Semantic Prior for Multi-Label Recognition With Incomplete Labels.
>
> Thank you for pointing this out. In the revised version, we have added new baselines, including MIME[1] and GR Loss[2], and the updated comparison results are shown in the table below. Our method outperforms these recent approaches on most datasets, with only a slight gap compared to the best method on CUB. Regarding *“Exploring Structured Semantic Prior…”* and other CLIP-based pseudo-labeling approaches, they introduce extra supervision information beyond single-positive labels, making direct comparison potentially unfair. However, we will explicitly discuss these methods in the Related Work section to provide a more comprehensive overview.
>
> | **Method** |       VOC        |       COCO       |       NUS        |       CUB        |
> | :--------: | :--------------: | :--------------: | :--------------: | :--------------: |
> |    MIME    |   89.20 ± 0.16   |   72.92 ± 0.26   |   48.74 ± 0.43   | **21.89 ± 0.35** |
> |  GR Loss   |   89.42 ± 0.23   |   73.15 ± 0.22   |   49.09 ± 0.02   |   21.36 ± 0.12   |
> |   CRISP    | **89.82 ± 0.19** | **74.64 ± 0.22** | **50.00 ± 0.32** |   21.65 ± 0.18   |
>
> [1] Chen, Yanxi, et al. "Boosting single positive multi-label classification with generalized robust loss." Proceedings of the Thirty-Third International Joint Conference on Artificial Intelligence. 2024.
>
> [2] Liu, Biao, et al. "Revisiting pseudo-label for single-positive multi-label learning." International Conference on Machine Learning. PMLR, 2023.
>
> For Questions:
>
> > 1. Will the inclusion of the additional absolute operator of Equation 7 compromise the unbias of the proposed risk estimator?
>
> The absolute value is introduced to ensure the non-negativity of the risk and to better align the model-predicted class prior with the ground-truth class prior. This modification does not compromise the unbiasedness of the estimator, as formally shown in Theorem 4.2. Despite the introduction of the absolute operator, we provide a estimation error bound demonstrating that the empirical risk minimizer still converges to the optimal risk minimizer as $n \to \infty$. The bound has already accounted for this modification.
>
> > 2. Why weren’t BOOSTLU and PLC compared on the MLL datasets?
>
> As we explained in lines 224-227 of our paper, due to the inability to compute the loss function of PLC without data augmentation, we do not report the results of PLC on MLL datasets because data augmentation techniques are not suitable for the MLL datasets. Similarly, since the operations of BOOSTLU for CAM are not applicable to the tabular data in MLL datasets, its results are also not reported.
>
> We hope that our revisions have addressed all of your concerns, but please let us know if there is anything else we can do to improve the manuscript. We would be happy to answer any additional questions or provide any further information you may need.

---

> > ### Comment · Reviewer_GgxX · 2025-08-05
> >
> > The responses partially addressed my concerns, and I would like to keep my original score.

---

### Note · Authors · 2025-08-12

We thank the Area Chair and reviewers for valuable feedback that improved the rigor, depth and clarity of our work. We are encouraged that our core contributions have been widely recognized, and through supplementary experiments and analyses, we believe we have comprehensively addressed all concerns raised.

## Strengths Acknowledged by Reviewers

• **Solid Theoretical Foundation** Reviewers (GgxX, wnkV) noted the theoretical guarantees for class-prior convergence and the consistent risk estimator that ensures convergence to optimal risk minimizer.

• **Comprehensive Experimental Validation** Reviewers (GgxX, 8tLt) praised the extensive experiments demonstrating necessity of addressing class-prior issues and effectiveness through ablation studies and time complexity analysis.

## Concerns & Responses

| Reviewer | Concerns                                                     | Responses                                                    |
| -------- | ------------------------------------------------------------ | ------------------------------------------------------------ |
| **GgxX** | (1) Method dependency on warm-up classifier quality (2) Missing recent baselines | (1) Conducted ablation studies showing robustness even with weak classifiers (2) Added MIME and GR Loss baselines, demonstrating superior performance on most datasets |
| **5pXo** | (1) Unclear method description and theoretical details (2) Questioned superiority over naive frequency estimation | (1) Clarified class-prior estimation process and sampling assumptions (2) Provided theoretical proof that naive frequency estimation is biased in SPMLL settings (3) Demonstrated empirically that naive methods degrade with limited annotations |
| **8tLt** | (1) Outdated baseline methods from several years ago (2) Limited evaluation metrics | (1) Added recent baselines (MIME, GR Loss) showing competitive performance (2) Clarified that multiple standard metrics were already used (mAP, Ranking Loss, Hamming Loss, One-Error, Coverage) |
| **wnkV** | (1) Method dependency on warm-up classifier quality (2) alternative class-prior estimation approach | (1) Demonstrated method robustness across different warm-up qualities through ablation studies (3) Showed comparison with validation-set frequency estimation in ablation study |

We sincerely thank all reviewers for their constructive feedback and are grateful to have received positive recognition from all reviewers for our work.

---

### Decision · Program_Chairs · 2025-09-17

**Decision:**

Accept (poster)

**Comment:**

This paper addresses the problem of single-positive multi-label classification, where every data point is only labeled with a single positive label. It proposes a novel approach to the problem, provides theoretical justification and demonstrates strong results. The rebuttal and discussion (especially the back-and-forth with reviewer 5pXo) played a big role in leading all reviewers to recommend acceptance.